# In situ analysis of nanoparticle soft corona and dynamic evolution

Didar Baimanov [1,2,3,4,5,9], Jing Wang [3,9], Jun Zhang[2,4], Ke Liu[1,5], Yalin Cong [1,4,5], Xiaomeng Shi[3], Xiaohui Zhang[3], Yufeng Li [1,5], Xiumin Li [1,5], Rongrong Qiao[1,5], Yuliang Zhao [1,4,6,7], Yunlong Zhou [2,4,8 ✉], Liming Wang [1,3,4,5 ✉] & Chunying Chen [1,4,6,7 ✉]

How soft corona, the protein corona's outer layer, contributes to biological identity of nanomaterials is largely because capturing protein composition of the soft corona in situ remains challenging. We herein develop an in situ Fishing method that can monitor the dynamic formation of protein corona on ultra-small chiral $Cu_2S$ nanoparticles (NPs) allowing us to directly separate and identify the corona protein composition. Our method detects spatiotemporal processes in the evolution of hard and soft coronas on chiral NPs, revealing subtle differences in NP − protein interactions even within several minutes. This study highlights the importance of in situ and dynamic analysis of soft/hard corona, provides insights into the role of soft corona in mediating biological responses of NPs, and offers a universal strategy to characterize soft corona to guide the rational design of biomedical nanomaterials.

Host proteins rapidly contact the surface of nanomaterials (NMs) in biological fluids and form an organic molecule layer known as the protein corona[1,2]. The composition of the corona confers biological identity to NMs and probably directs further interactions with proteins, cells, tissues, etc.[3,4]. The formation and evolution of protein corona are thermodynamic and kinetics processes, during which dynamic exchange of corona components is based on different abundance and diffusion rates of proteins, and varying affinities of different proteins binding to the NMs. Proteins exhibiting high binding affinity and slow dissociation rate form the hard corona (HC), while ones with low binding affinity and fast dissociation rate form the soft corona (SC)[5–7]. Previous studies found that the HC and SC play distinct roles in cellular uptake and inflammatory responses to NMs, specifically in cases of protein-precoated NMs and NMs in serum-containing medium[8,9]. To address the nature of NM-mediated biological behaviors and responses[10–12], it is vital to understand the role of HC/SC composition and the effects of NMs' physicochemical properties, with the ultimate goal of dynamically changing the formation, evolution, and function of protein coronas in biological systems. A longstanding question remains regarding how the corona components mediate the recognition and binding of NMs to the cell surface and subsequent biological responses. One major reason is the difficulty in obtaining the actual composition of SCs in situ, as these loosely attached proteins are often missed during corona sample separation with repeated high-speed centrifugation and continuous elution[13,14]. Therefore, a real-time and in situ strategy for monitoring and manipulation of protein coronas on the surface of NMs is in high demand, as this development would allow direct study of the formation and evolution of HCs/SCs

[1]CAS Key Laboratory for Biomedical Effects of Nanomaterials and Nanosafety, CAS Center for Excellence in Nanoscience, Institute of High Energy Physics & National Center for Nanoscience and Technology of China, Chinese Academy of Sciences, Beijing 100049, P. R. China. [2]Zhejiang Engineering Research Center for Tissue Repair Materials, Wenzhou Institute, University of Chinese Academy of Sciences, Wenzhou 325000 Zhejiang, P. R. China. [3]State Key Laboratory of Natural and Biomimetic Drugs, School of Pharmaceutical Sciences, Peking University, Beijing 100191, P. R. China. [4]University of Chinese Academy of Sciences, Beijing 100049, P. R. China. [5]CAS-HKU Joint Laboratory of Metallomics on Health and Environment & National Consortium for Excellence in Metallomics, Institute of High Energy Physics, Chinese Academy of Sciences, Beijing 100049, P. R. China. [6]GBA Research Innovation Institute for Nanotechnology, Guangzhou 510700 Guangdong, P. R. China. [7]Research Unit of Nanoscience and Technology, Chinese Academy of Medical Sciences, Beijing 100730, P. R. China. [8]Oujiang Laboratory, Zhejiang Laboratory for Regenerative Medicine, Vision and Brain Health, Wenzhou 325001 Zhejiang, P. R. China. [9]These authors contributed equally: Didar Baimanov, Jing Wang. ✉e-mail: zhouyl@ucas.ac.cn; wangliming@ihep.ac.cn; chenchy@nanoctr.cn

and accurate identification of their compositions in physiological fluids.

To characterize SC components, several analysis methods have been developed. For example, cryo-transmission electron microscopy (cryo-TEM) is a powerful technique for observation of the loose network of SCs formed on the surface of nanoparticles (NPs)[15,16]. Based on a lock-in filtering algorithm, a real-time 3D single-particle-tracking spectroscopy can be used to study SC formation and exchange by model proteins on single NPs[17]. Noticeably, an automated platform based on magnetic NMs can be used to rapidly perform plasma proteome profiling to study protein coronas without centrifugation[18]. Using a click-chemistry-based strategy, weakly bound proteins are locked by cross-linking with HC proteins, centrifugated, and eluted for MS-based compositional profiling of the SC proteins[19]. Moreover, anti-PEG single-chain variable fragment (PEG-scFv)-based affinity chromatography (AfC) can separate and identify total protein corona on the PEGylated liposomes[20]. Asymmetric flow field-flow fractionation (AF4) was also used to separate SC proteins from the surface of NMs[21]. Underlying a major challenge for SC separation and identification, current methods primarily rely on centrifugation separation, which requires the isolation of NM−corona complexes from biological fluids via high-speed centrifugation and buffer rinses. However, only HC proteins are retained during centrifugation, with SC components lost. For accurate analysis of SC composition, it is crucial to monitor the real-time formation of SC in physiological fluids such as blood, plasma, and serum, as well as to manipulate and separate composite proteins in situ for liquid chromatography–tandem mass spectrometry (LC−MS/MS) identification in complex systems, allowing us to analyze time- and surface-dependent protein corona formation and evolution.

Chirality is an essential natural characteristic and plays a major role in the biological and physiological processes of living organisms[22,23]. Due to the high selectivity for chiral biomolecules exhibited by biological systems[24], chiral nanotechnology has become a popular research field over the last decade[25]. Chiral NMs have been identified as promising functional materials for multiple applications, including disease therapy, pathogen prevention, disease diagnosis, and detection of biological molecules in biomedical fields[26–28]. Moreover, chiral NMs have been found to induce distinct biological responses, including uptake, autophagy, energy metabolism, and cell differentiation, due to their stereo-selective surfaces[29–33]. To apply chiral NMs in vivo, it is of vital importance to know whether NM chirality influences performance or trafficking processes such as adsorption, distribution, metabolism, and excretion (ADME). The study of soft corona composition on the NM surface can help predict and explain differences in stereo-selective behaviors, fates, and biological responses of NMs. As a new class of semiconductor material, copper sulfide NPs are attracting increasing attention in the biomedical field due to their unique optical properties, degradability, and chemical activity[34–37]. We thus chose ultra-small chiral cuprous sulfide ($Cu_2S$) NPs as a model NM to investigate the dynamic formation and evolution of SC and HC on the chiral surface and tried to correlate SC to cellular uptake by macrophages, blood clearance, and tissue accumulation, and metabolism behaviors in vivo.

To resolve the challenges of SC analysis, we herein develop fast and bio-layer interferometry (BLI)-based Fishing strategy based on the immobilization of chiral $Cu_2S$ NPs on biosensors, adsorption of proteins, and controllable elution of SC proteins coupled with LC−MS/MS for protein identification. Compared to the centrifugation-based method, this integrated strategy is powerful to study the formation and evolution of protein corona that allows real-time monitoring of protein adsorption and dissociation[38–40] and supports in situ and fast isolation of multiple layer coronae, i.e., HC and SC for accurate identification and quantification of protein components. The BLI-based Fishing method is also feasible to perform corona study in complex biological fluids without further centrifugation and in dynamic and very fast interaction systems for protein corona especially for ultrasmall NPs, ultrathin nanosheets, and NMs with low density that is hard to be centrifugated in seconds and several minutes. Based on this strategy, we herein investigated SC formation over time for two L/D-cysteine-capped chiral $Cu_2S$ NPs in a 10% mouse serum-containing buffer as a model system. We found that the protein composition in both HC and SC on the two chiral NPs showed an obvious difference; for both chiral NPs, the difference in HC composition increased with time, while the difference in SC composition decreased with time after 3 and 30 min of incubation. The results suggested that the stereo-selective abundance of HC and SC composition was distinct over incubation time. Interestingly, the differences in soft corona components by chirality may explain the differences in blood clearance rates for two chiral NPs in vivo and in cellular uptake rates in vitro. This study well correlates in vivo fate of chiral NPs with their soft coronal components. The present study emphasizes the promising application of this method for real-time, in situ, and accurate analysis of SC supports the broad utilization of soft corona analysis for various nanomaterials within complex biological environments, and promotes the rational design of functional nanomedicines.

## Results and discussion

### Physicochemical properties of chiral cysteine-capped $Cu_2S$ NPs

Two ultrasmall and chiral cysteine-capped $Cu_2S$ NPs (D-/L-NPs, Fig. 1a, b) were initially synthesized (Supplementary Fig. 1a), which were characterized, including size, morphology, surface, and structural properties, using the techniques of transmission electron microscopy (TEM), X-ray photoelectron spectroscopy (XPS), zeta potential, circular dichroism (CD) spectra and X-ray diffraction (XRD) (Fig. 1c–h and Supplementary Fig. 1). TEM revealed the spherical shape and monodisperse nature of the D-/L-NPs with average sizes of $4.2 \pm 0.5$ and $4.1 \pm 0.7$ nm (Fig. 1c, d), respectively. In addition, XPS analysis was used to perform elemental identification and assess the valence state of NPs (Fig. 1e, f) as a $Cu_2S$ form. The surface charges of D-/L-NPs in PBS were −33.3 and −33.8 mV, respectively (Fig. 1g). CD spectra showed a symmetrical intensity with different chirality due to the D- and L-cysteine templates (Fig. 1h). In summary, we synthesized two chiral NPs with approximately identical physicochemical properties, including size and surface charge, but different surface chirality.

### Isolation of protein corona components by a Fishing method

We developed a Fishing strategy to dynamically characterize two types of coronas (i.e., HC and SC), allowing accurate and fast isolation of proteins from the surface of NMs without further centrifugation of samples. Protein corona formation and evolution is a spatiotemporal process, during which both NP−corona complex structure on the outer and inner layers and its dynamic exchange with incubation time make it difficult to precisely define hard and soft corona. To better address the challenges, we divided the multilayers of corona into outer and inner sections, i.e., the soft and hard coronas in this study. To study corona formation on the surface of ultra-small $Cu_2S$ NPs in real-time, we loaded two chiral NPs (L-NPs and D-NPs) onto immunoglobulin G (IgG)-immobilized amine-reactive 2nd generation (AR2G) biosensors, which are able to monitor dynamical interactions between NPs and serum proteins (Fig. 1i). Specifically, IgG was first immobilized on the AR2G biosensors through the amine-coupling method (refer to the "Methods" section for details). Then, the chiral NPs were loaded onto the IgG-coated biosensor due to the high binding affinity of IgG to NPs. Because of this affinity, the chiral NPs remain on the sensor after washing with 0.1% washing buffer (WB) (Supplementary Fig. 2). Subsequently, chiral NP-loaded biosensors were immersed in 10% mouse serum to allow protein corona formation for 3 and 30 min under continuous shaking of microplate at 37 °C. Rapid adsorption of serum proteins was monitored in real-time as reflected by an increase in the wavelength shift ($\Delta\lambda$) of the biosensor. When proteins adsorb on the

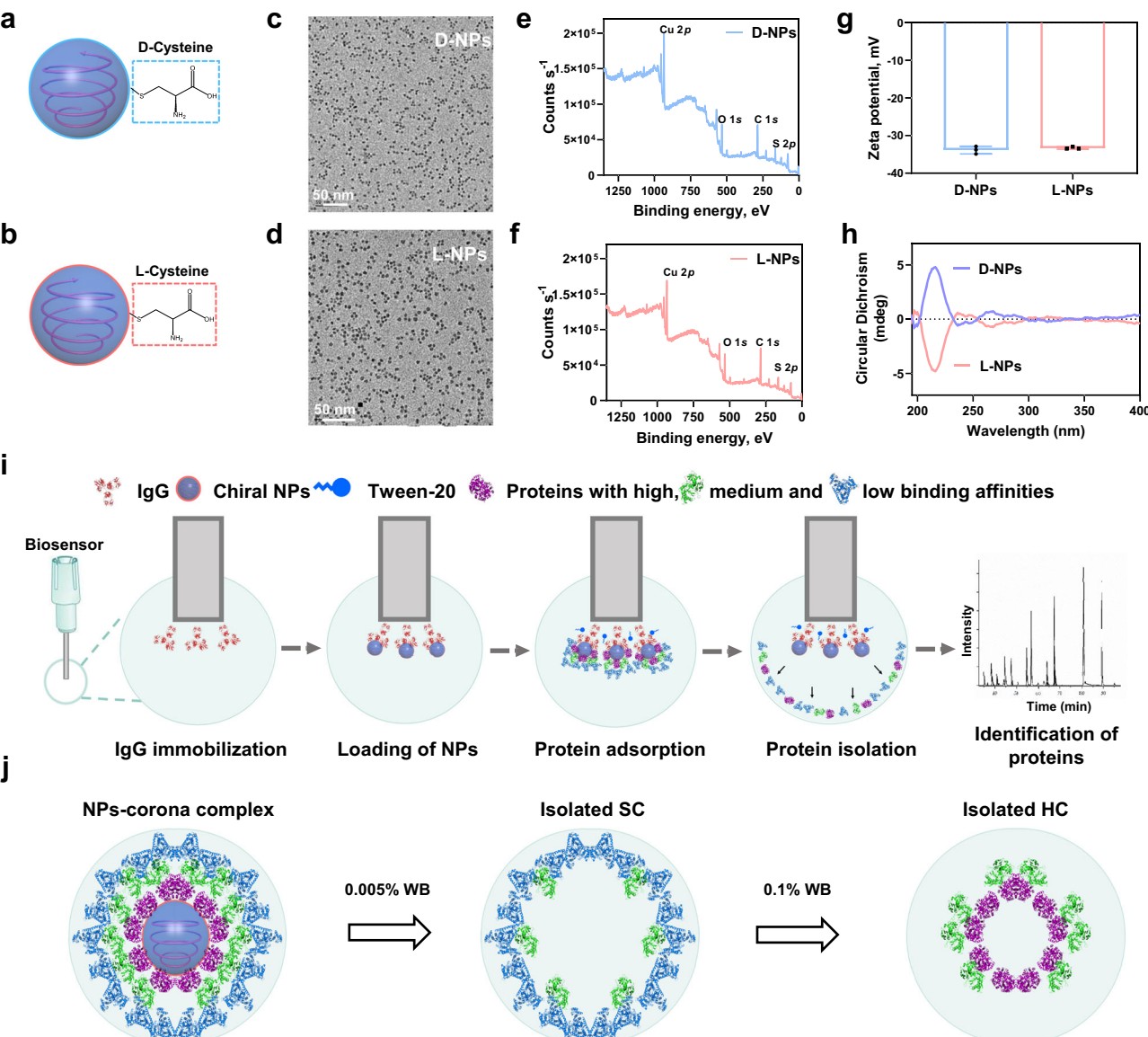

**Fig. 1 | Characterization of chiral Cu₂S NPs and the workflow of BLI-based Fishing method. a, b** Structure schemes of ᴅ-cysteine (the circle with blue color) templated Cu₂S NPs (ᴅ-NPs) and ʟ-cysteine (the circle with red color) templated Cu₂S NPs (ʟ-NPs). **c–h** Characterization for the size/shape, electronic structures, surface charges, and chirality of ᴅ-NPs and ʟ-NPs by TEM (**c, d**), XPS (**e, f**), zeta potential (**g**), and circular dichroism (**h**). The scale bar represents 50 nm. **i** Schematic illustration of BLI-based Fishing strategy including protein corona formation on the chiral NPs, the isolation of corona in situ, and identification of protein composition and abundance. The whole method includes five steps: (1) the coating of the biosensors with IgG; (2) the loading of chiral NPs onto the biosensors;

(3) the incubation of NPs with biological fluids and the formation of protein corona; (4) isolation of protein coronas from the NPs; (5) protein digestion and proteomic analysis for identification of corona components by LC–MS/MS. The spherical structures with gray blue and blue colors refer to Cu₂S NPs and Tween-20, respectively; while the irregular structures with purple, green, light blue, and red colors represent various types of proteins. **j** Schemes demonstrating the isolation of soft corona (SC) and the hard corona (HC) in two steps by washing buffer (WB). All data are shown as mean values and standard deviations for triplicate biological samples (*n* = 3). Source data are provided as a Source Data file.

bottom of an NP-loaded BLI biosensor and form a bio-layer, the change in bio-layer thickness results in a shift in light wave interference (Supplementary Fig. 2), which can be used to sensitively quantify the adsorbates according to interferogram shift[40].

To optimize the Fishing procedure, we tested various concentrations of WB (Supplementary Fig. 2) to ensure efficient washout of the protein corona and elution from the NP-loaded biosensors. Pure water was first used as a control buffer, for which proteins adsorbed on the NPs were not washed out based on BLI results. Next, we observed that proteins on the NPs could be eluted by WB in a concentration-dependent manner from 0.005% to 0.1% (w/w). Specifically, treatment with 0.005% (w/w) WB led to partial dissociation (-30%) of coronal

proteins, as determined by bio-layer wavelength shift. In comparison, treatment with 0.1% WB could detach nearly 100% of coronal proteins from NP-loaded biosensors. Afterward, the multilayers of adsorbed proteins, namely, the soft and hard coronas, were in sequence and isolated by the Fishing method (Fig. 1j and Supplementary Fig. 3) for further LC–MS/MS characterization. To clarify corona composition and organization, isolated proteins with weak binding affinity at the outer layer are described as SC, and proteins with stronger binding affinity at the inner layer are described as HC. Therefore, WB at present concentrations could elute multilayered proteins from the surface of NPs with the immobilized NPs on the sensors during the washing (Supplementary Figs. 2 and 3).

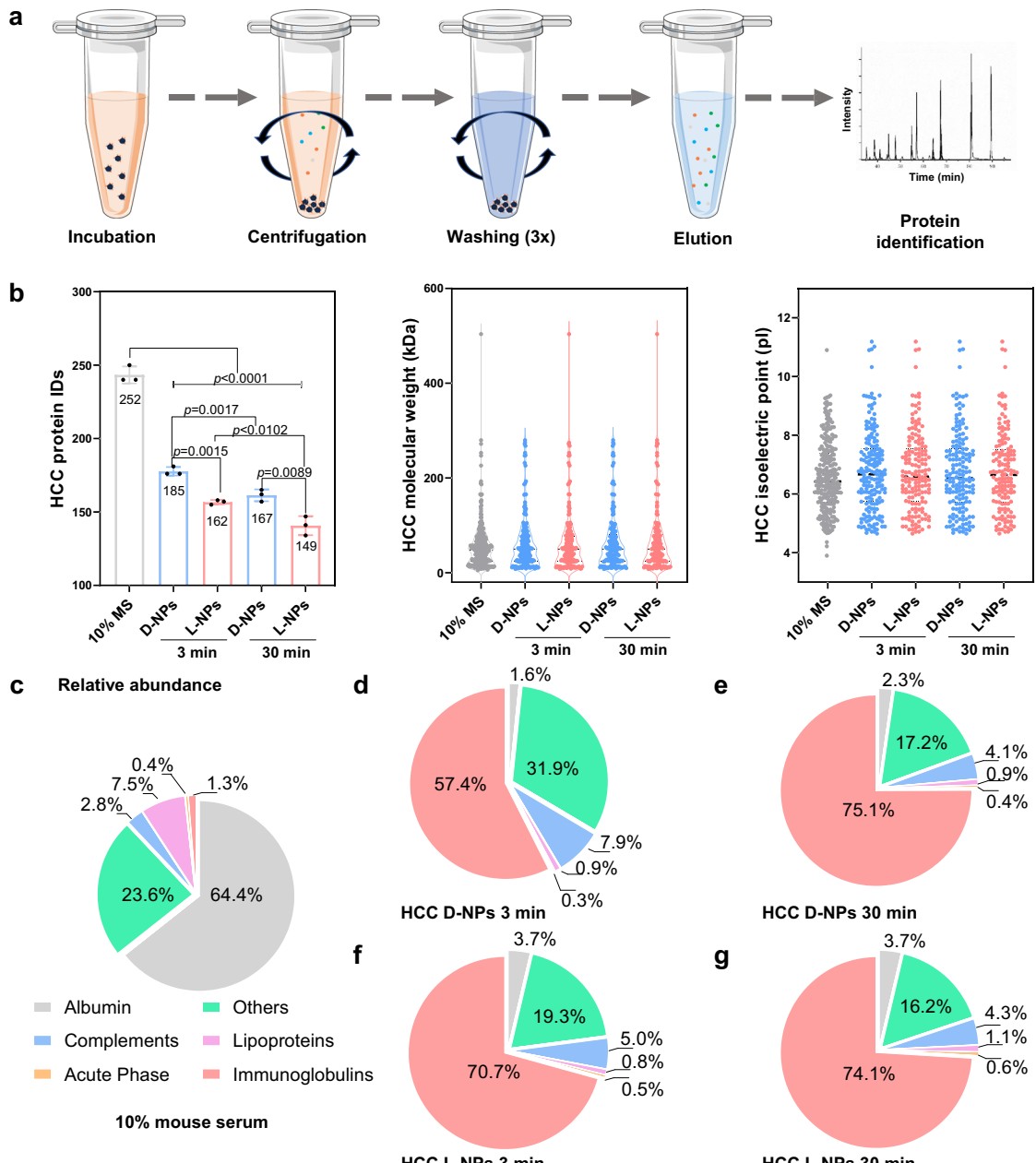

**Fig. 2 | Proteomic characterization of the hard corona components by centrifugation (HCC) on chiral NPs. a** Schematic illustration of HCC isolation according to centrifugation. The incubation solution is shown as orange, and WB before and after elution is shown as gray blue and light blue, respectively. NPs are shown as black and various types of proteins are shown as dots with blue, red, and green colors. **b** Proteomic information for composition (IDs), molecular weight, and isoelectric point of 10% mouse serum (MS) proteins and the identified HCC proteins on the chiral NPs upon 3 and 30 min incubation. All of the data are shown as mean values and standard deviations for triplicate biological samples (*n* = 3). **c** The protein composition and the abundance in 10% mouse serum were determined by nano LC–MS/MS and Proteome Discoverer 2.4. **d–g** Time-dependent

change in the components of HCC proteins on the chiral NPs classified by biological function. Pie-chart depicts the relative abundance of identified proteins belonging to each of five categories, including complement components, immunoglobulins, acute phase, lipoproteins, and other types. Comparably, with the highest abundance in the serum, albumin is classified as an individual category. Protein compositions are identified by manual searching of the Mouse Database via the UniProt website. Statistical significance was calculated by one-way ANOVA with Tukey's multiple comparisons test. *$p < 0.05$; **$p < 0.01$; ***$p < 0.001$; ****$p < 0.0001$; n.s., not significant ($p > 0.05$). Proteomics data are presented in Supplementary Data 1. Source data are provided as a Source Data file.

## Identification of hard corona proteins by centrifugation

To assess the differences in HC composition between the centrifugation approach and the Fishing approach, we first isolated hard corona proteins by centrifugation (HCC) of two chiral NPs after 3 and 30 min incubation with 10% mouse serum at 37 °C (Fig. 2a)[8,41]. Then, the suspension of NPs with 10% mouse serum was centrifuged at 360,000×*g* for 1 h to remove free or loosely bound proteins and then to obtain the precipitation containing stable hard corona–NP

complexes, HCC. HCC proteins were further collected and digested for LC–MS/MS analysis.

Based on the LC–MS/MS results, we identified 252 different types of proteins in 10% mouse serum and 185, 167 and 162, 149 types of proteins on the surface coronas of ᴅ-/ʟ-NPs after 3 and 30 min incubation (Fig. 2b), respectively. To further correlate NP chirality with HCC composition, the complete data sets, including molecular weight, calculated isoelectric point, and protein classifications associated with

biological processes, of identified serum and HCC proteins on D-/L-NPs were analyzed, as shown in Fig. 2b–g and Supplementary Fig. 4, and a list of the top 20 most-abundant proteins for each sample was generated (Supplementary Data 1). In 10% mouse serum, albumin is the most abundant protein, with a percentage of 64.4%, followed by others, lipoproteins, complements, immunoglobulins, and acute phase proteins (Fig. 2c). Compared to serum, the relative abundance of albumin decreased to 1.6%, 2.3% and 3.7%, 3.7% on the surface of D-/L-NPs upon 3 and 30 min incubation, respectively. Noticeably, the relative abundance of immunoglobulins significantly increased from 1.3% in mouse serum to ~57.4%, ~75.1% and ~70.7%, ~74.1% on the surface of D-/L-NPs upon 3 and 30 min incubation, respectively. Followed by the decreased abundance of others and lipoproteins, the abundance of complements increased in the HCC of chiral NPs (Fig. 2d–g). Similar to our results, previous work showed that the IgG abundance increased, and HSA decreased after multiple washing steps[42]. Thus, both HCC composition and abundance on two chiral NPs differed from those in the serum (Supplementary Fig. 4).

Furthermore, we observed that a high similarity in HCC protein types between D-NPs and L-NPs was observed over time (Supplementary Fig. 5), but HCC components exhibited significantly different with time (Supplementary Fig. 4), suggesting that HCC is influenced by the surface chirality of NPs (i.e., the formation of the chirality-specific hard corona). In addition, the results of unsupervised hierarchical clustering demonstrated differential enrichment of serum proteins on chiral NPs (Supplementary Fig. 6), suggesting the effect of chirality on HCC composition and components.

### Time-dependent change in HC composition for chiral NPs

When performing centrifugation-based HC preparation, the outer layer of adsorbed proteins (i.e., the SC) could be overlooked. A BLI-based Fishing method was thus used to perform in situ separation of the corona–NP complex from 10% mouse serum, after which SC and HC proteins (Fig. 1j) were eluted from the complex. The HC isolation and collection protocol are shown in Fig. 3a and Supplementary Fig. 3. HC composition significantly differed from HCC composition (Supplementary Fig. 6). Interestingly, protein composition showed low similarities (<30%) between HCC and HC over time for both chiral NPs (Supplementary Fig. 7) due to different ways to separate NP–protein complex, i.e., the centrifugation and rinsing steps required for HCC, versus in situ isolation for HC. Although the protein corona formation runs under similar conditions, isolation steps inevitably affect the result of coronal composition and abundance.

We found that HC composition did not change obviously over time for both NPs, during which the number of HC protein species changed from 59 to 60 for D-NPs and from 66 to 69 for L-NPs from 3 to 30 min (Fig. 3b). The number of HC protein species was much smaller than that of HCC for D-NPs (185 vs.167) and L-NPs (162 vs.149) after 3 vs. 30 min. Detailed information regarding HC proteins on D-/L-NPs at different incubation times was provided, including molecular weight, calculated isoelectric point, and protein classifications associated with biological processes (Fig. 3b and Supplementary Data 1). Compared to centrifugation-based methods, the Fishing method supports fast and easy separation of HC proteins, allowing us to perform time-dependent analysis of protein adsorption and exchange.

To understand the time-dependent evolution of HC, we analyzed quantitative changes in HC composition and abundance after 3 and 30 min incubation. Based on BLI adsorption curves, we observed rapid adsorption of serum proteins onto NPs even at 3 min of incubation (Supplementary Fig. 3). The abundance of HC composition significantly changed over time although the protein composition on chiral NPs was similar at 3 and 30 min (Fig. 3c, d). This might be caused by the rapid and chaotic adsorption of proteins on the NPs at the early stage, followed by the binding equilibration[43,44]. Protein exchange was more consistent over longer interaction times, suggesting that the more rapid exchange observed on shorter time scales is due to competitive displacement[45]. Classified by biological functions, the proteins identified on chiral NPs were grouped and ordered by relative abundance for D-/L-NPs (Fig. 3c, d and Supplementary Figs. 8, 9). A list of the top 20 most abundant HC proteins on D-NPs and L-NPs based on three replicates is shown (Supplementary Data 1). At both time points, a major portion of HC proteins for chiral NPs could be classified as other proteins, followed by albumin. With increasing incubation time, the abundance of immunoglobulins and acute phase proteins increased for both chiral NPs, while those of albumin and other proteins decreased. Among all types of proteins, the abundance of immunoglobulin and acute phase proteins increased the most significantly (Fig. 3c, Supplementary Figs. 8, 9).

We next studied the effect of surface chirality on HC properties, including composition and abundance. After 3 min of incubation (Fig. 3c, Supplementary Fig. 8), the relative abundance of acute phase proteins on both chiral NPs showed a significant difference (~0.8% for D-NPs vs. ~16% for L-NPs). Moreover, the chirality-specific difference in HC protein abundance was ~6.2% for albumin and ~10.5% for other proteins. Interestingly, after 30 min incubation (Fig. 3c, Supplementary Fig. 9), a significant difference in HC protein abundance on both NPs was observed for albumin (~6.1%), others (~16.1%), acute phase (~9.8%), and immunoglobulin (~11.0%). A similar trend in HC abundance for chiral NPs was that albumin and other proteins decreased, while acute phase and immunoglobulins increased over time. Collectively, chirality influences HC composition and abundance at the early stage with significant changes over time (Supplementary Fig. 10a, b), highlighting the dynamic effect of chirality (Supplementary Fig. 10c, d) on HC formation, i.e., dynamic fingerprints for chiral NPs' HC. Herein, we elucidate that both time and surface chirality of NPs play important roles in the formation and evolution of hard corona.

### Dynamic evolution of SC composition

We further used the BLI-based Fishing method to investigate the dynamic evolution of weakly bound SC proteins on chiral NPs (Fig. 4a) and compared HC and SC properties. Molecular weight, calculated isoelectric point (Fig. 4b and Supplementary Data 1), and protein classifications by biological processes for identified SC proteins (Supplementary Figs. 8, 9) are provided. A list of the top 20 most abundant proteins at 3 and 30 min (Supplementary Data 1) was generated. High similarity in coronal composition between HC and SC was found on two chiral NPs (>68%) at 3 and 30 min (Supplementary Fig. 10e–h). Although the same proteins can simultaneously exist as soft and hard corona compositions, the abundance of HC and SC proteins was obviously different at 3 and 30 min (Supplementary Fig. 11a, b). The reason was that protein adsorption and exchange are thermodynamic and kinetics processes. At the beginning of adsorption, the proteins with high abundance but low diffusion rates can be immediately adsorbed on the surface of NPs as SC or the inner layer, in which such a thermodynamic process is influenced by protein abundance and diffusion rates. With the prolonging time, the competitive adsorption of various proteins with higher binding affinities to the surface of NPs drives the exchange of protein composition and abundance at the inner layers, which is a kinetics process for protein adsorption. Thus, the formation and evolution of protein corona is not only time-dependent interaction, but also a dynamic change in spatial structures of corona including its organization of the outer and inner layers, multiple compositions, and different abundance of proteins.

Overall, at 3 min incubation, the abundances of immunoglobulins, lipoproteins, and complements in HC and SC are similar on the surface of D-NPs, while those of albumin and other proteins are much higher in HC than SC and acute phase proteins are 26% vs. ~0.8% for SC vs. HC of D-NPs. As for HC and SC of L-NPs at 3 min, the amounts of acute phase proteins are about 16% and 0.7%, while immunoglobulins are about

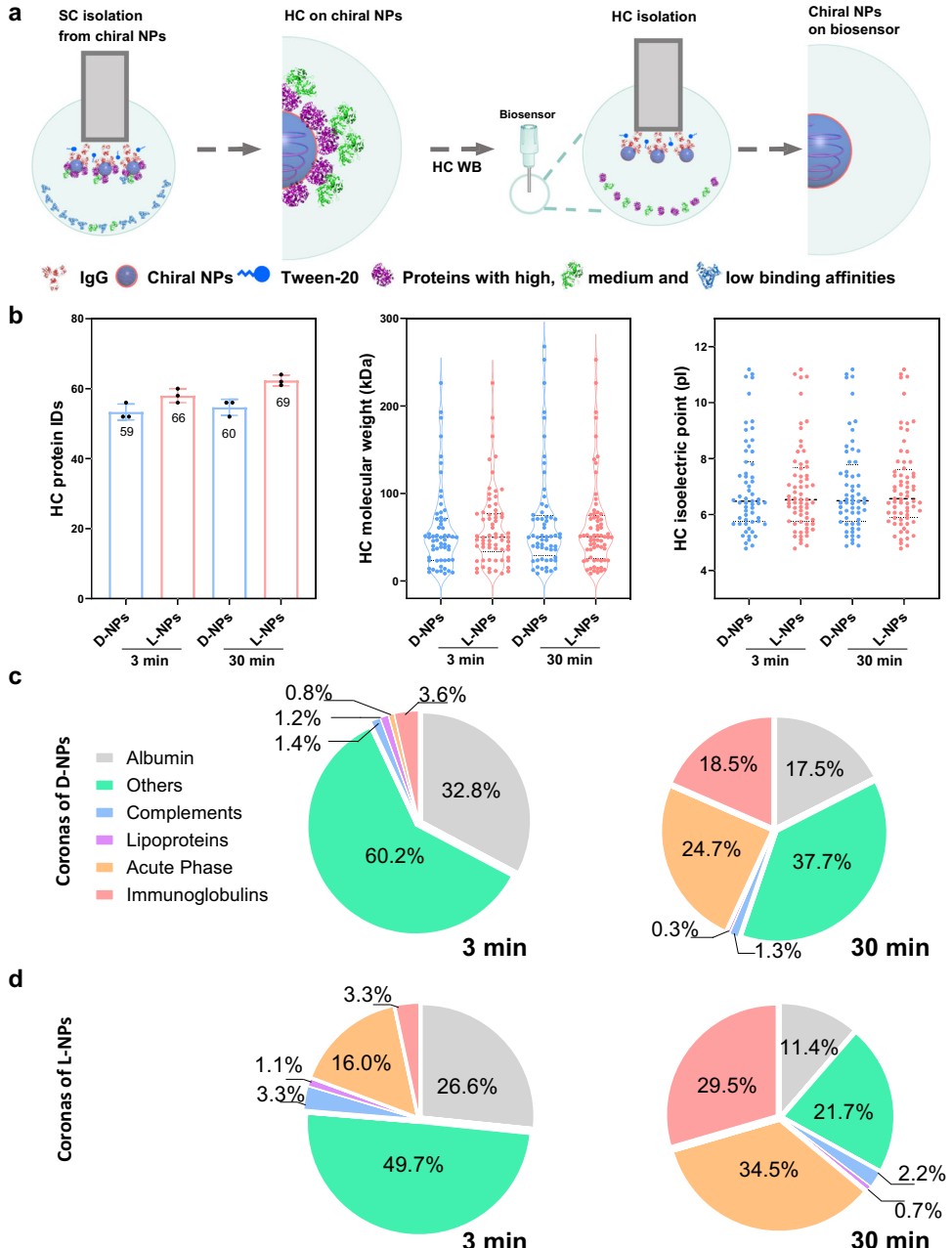

**Fig. 3 | Proteomic characterization of time-dependent hard corona (HC) components on chiral NPs. a** Schematic illustration of HC protein isolation from the surface of NPs according to BLI-based Fishing method. Multilayered protein corona immediately forms on chiral NPs loaded on the biosensors and then the outer layer, i.e., soft corona (SC) is eluted by SC washing buffer from protein coronal structure. Finally, the inner layer of the corona, i.e. hard corona can be isolated from the chiral NPs by HC washing buffer (WB). The spherical structures with gray blue and blue colors refer to Cu$_2$S NPs and Tween-20, respectively; while the structures with purple, green, light blue, and red colors represent various types of proteins. **b** Proteomic information for composition (IDs), molecular weight, and isoelectric point of the identified HC proteins on the chiral NPs upon 3 and 30 min interaction.

All of the data are shown as mean values and standard deviations for triplicate biological samples ($n = 3$). **c, d** Time-dependent change in the components of HC proteins on D-NPs and L-NPs classified by biological function. Pie-chart depicts the relative abundance of identified proteins belonging to each of five categories, including complement components, immunoglobulins, acute phase, lipoproteins, and other types. Comparably, with the highest abundance in the serum, albumin is classified as an individual category. Protein compositions are identified by manual searching of the Mouse Database via the UniProt website. Statistical significance is calculated by one-way ANOVA with Tukey's multiple comparisons tests; n.s., not significant ($p > 0.05$). Proteomics data are presented in Supplementary Data 1. Source data are provided as a Source Data file.

3.3% and 17.7%, respectively. Subsequently, we continued to observe for 30 min and found the persistent enriched level of acute phase proteins on SC and HC on both chiral NPs, while levels of other proteins decreased. Noticeable, the significant difference in immunoglobulins of SC on D-NPs was higher compared to L-NPs (5.8-fold). Our results indicated the formation of stereospecific protein corona at an early stage upon incubation with mouse serum, which was consistent

with a previous report that HC and SC have similar protein composition but at different quantities[19].

To further elucidate the time-dependent evolution of the SC, we compared the quantitative changes in SC composition and abundance between 3 and 30 min of incubation. Similar to the HC results, the number of SC protein species did not significantly change with time (Fig. 4b). We observed that 67% and 69% of SC components on D-NPs

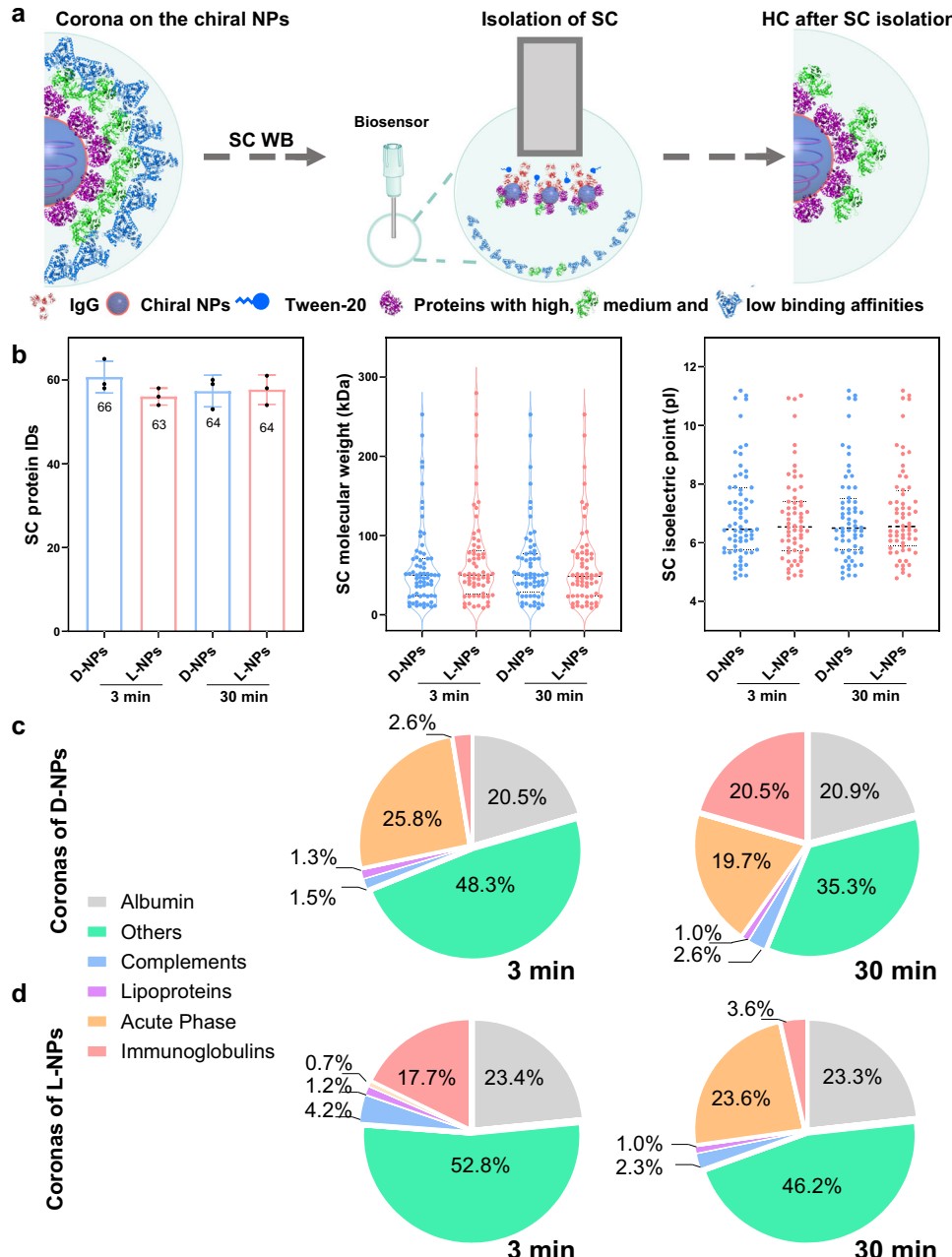

**Fig. 4 | Proteomic characterization of time-dependent soft corona (SC) components on chiral NPs.** **a** Schematic illustration of SC protein isolation from the surface of NPs according to BLI-based Fishing method. The spherical structures with gray blue and blue colors refer to $Cu_2S$ NPs and Tween-20, respectively; while the structures with red, green, light blue, and purple colors represent various types of proteins. **b** Proteomic information for composition (IDs), molecular weight, and isoelectric point of the identified SC proteins on the chiral NPs at 3 and 30 min. All of the data are shown as mean values and standard deviations for triplicate biological samples ($n = 3$). **c**, **d** Time-dependent change in the components of SC proteins on D-NPs and L-NPs classified by biological function. Pie-chart depicts the relative abundance of identified proteins belonging to each of five categories, including complement components, immunoglobulins, acute phase, lipoproteins, and other types. Comparably, with the highest abundance in the serum, albumin is classified as an individual category. Protein compositions are identified by manual searching of the Mouse Database via the UniProt website. Statistical significance is calculated by one-way ANOVA with Tukey's multiple comparisons tests; n.s., not significant ($p > 0.05$). Proteomics data are presented in Supplementary Data 1. Source data are provided as a Source Data file.

and L-NPs after 30 min interaction, were retained for those at 3 min (Supplementary Fig. 10i, j). Furthermore, we found that SC protein abundance varied with time and chirality. A major portion of SC proteins after 3 min could be classified as others (such as lumican and ADF-H domain-containing protein etc.), acute phase, and albumin for D-NPs, while as others, albumin and immunoglobulins for L-NPs (Fig. 4c, d). Interestingly, after 30 min, most SC proteins on D-NPs were classified as albumin (~20.9%), others (~35.3%), immunoglobulins (~20.6%), and acute phase proteins (~19.7%). With the increasing incubation time, the

abundances of albumin (~23.3%), other (~46.2%), and acute phase proteins (~23.6%) were classified as the most abundant proteins on L-NPs. Among all types of protein, the abundance of immunoglobulins and acute phase proteins changed most significantly for both chiral NPs (Fig. 4c, d). The abundance of apolipoproteins (as lipoproteins) on both chiral NPs did not change obviously with time. Understanding the impact of NPs' surface properties[46] and time-dependent changes on corona composition may better elucidate the nature of dynamic corona formation and biological responses to NPs. Our results suggest

that SC evolution is a complex and dynamic process that involves continuous exchange and the replacement of proteins with highly abundant proteins. We observed a correlation between the number of protein species and the relative abundance of persistent proteins for both chiral NPs over time.

We next studied the effect of chirality on SC evolution and properties and found that surface chirality caused slight variation in the number of SC protein species and the types. Specifically, there was no significant difference in the number of SC protein species between D-NPs and L-NPs at 3 and 30 min. Additionally, the similarity in SC composition for D-NPs and L-NPs overlapped by 61% and 71% at 3 and 30 min, respectively, suggesting slight differences in SC protein types (Supplementary Fig. 10k, l), however, the abundance of SC proteins on L-NPs and D-NPs was different at 3 and 30 min (Supplementary Fig. 11c, d). Furthermore, we compared levels of highly abundant proteins on the chiral NPs. Albumin was found as one of the major SC components for both chiral NPs and its abundance did not change obviously with time and surface chirality (Fig. 4c, d). Noticeably, other proteins could be gradually replaced with immunoglobulins and acute phase proteins over time. This result is consistent with a previous study that albumin was a major component of the SC on the surface of polystyrene NPs[21]. Surprisingly, we observed that abundance of the immunoglobulin (Mab 110 heavy chain) only presented in the SC at 3 min and HC at 30 min on L-NPs, suggesting significant re-modeling of protein corona over time. With respect to D-NPs, this protein was detected on both SC and HC only at 30 min, suggesting the stereoselective recognition of chiral NPs over time (Supplementary Data 1). Although chirality resulted in slight differences in the number of SC protein species, the coronal composition and abundance were noticeably different on the surface of chiral NPs. Therefore, proteins located in the outer layer of corona serve as a fingerprint of the NPs and may play roles in recognition by the biological systems, including cells, tissues, and organisms. We thus demonstrated that surface chirality, incubation time, and HC composition (protein–protein interaction) are crucial factors in determining SC properties.

Interestingly, albumin remained a major component of SC, due to its high abundance in serum. As a result, albumin in SCs remained similar in abundance for both chiral NPs over time (Fig. 4c, d), while albumin in HCs dramatically decreased by about 15% (Fig. 3c, d). With respect to immunoglobulins, the protein enrichment level (the abundance of a protein on NPs compared to that in the serum) on D-NPs increased with time for both HC and SC and its abundance on HC and SC was similar at two-time points. Comparably, the enrichment level of immunoglobulins on L-NPs increased with time for HC, however, the level decreased with time for SC. The results suggested that L-NPs and D-NPs showed the same trend for the enrichment of immunoglobulins in HC, but they exhibited the opposite trend for immunoglobulin in SC.

### Distinct binding affinity of proteins to NPs and formed HC

To understand SC and HC formation on the surface of L-NPs, we next investigated the association and disassociation of specific proteins on NPs and mouse serum-precoated NPs during SC formation. We assessed two types of representative and abundant proteins, albumin and IgG (representing immunoglobulins), as examples to study NP–protein and protein–protein interactions. After loading with L-NPs, sensors were immersed in 10% mouse serum for 3 min and rinsed with water to form a coronal layer. The abundances of albumin and immunoglobulins in 10% mouse serum were 64.4% and 1.3%. For the HC at 3 min, the percentages of these proteins on L-NPs were 26.6%, and 3.3%, respectively (Fig. 3d). After the incubation with mouse serum, immunoglobulins were enriched in HC, however, the percentage of albumin decreased obviously (Supplementary Fig. 12). To explain the chirality-specific protein enrichment in coronas, BLI results showed that these proteins have strong binding affinities ($K_D < 10$ nM) with L-NPs, suggesting that these proteins might contribute to the HC.

Specifically, IgG and HSA exhibited strong binding affinities ($K_D < 1$ pM and ~9.9 nM) with L-NPs (Fig. 5a-c). These results indicated that IgG and HSA can be enriched in HC, in particular, IgG. The composition of HC formed on NPs is thus largely determined by the binding affinities of different HC components to NPs.

Next, we further studied SC formation on the pre-formed corona layer on NPs in order to explain why albumin is the major component of SC at the single protein level. According to BLI results (Fig. 5d–f), the binding constant of IgG (with a $K_D$ value of ~1.2 nM) to the pre-formed corona layer was 27-fold less than those of albumin (with a $K_D$ value of 32.5 nM). These results indicated that IgG can be highly enriched in the SC layer compared to albumin. Furthermore, we attempted to understand how the competitive interaction of neighboring proteins might impact the corona formation (Fig. 5g–i). We observed that IgG could be strongly bound to the surface of L-NPs coated by IgG and HSA, however, the HSA showed much weaker binding. These results indicated that binding affinity alone cannot fully address the respective contributions of these proteins to the SC. High abundance in the serum is another important factor that may explain why albumin is the major component of SC.

### Correlation of chiral surface with biodistribution of NPs

We next examined whether surface chirality can modulate in vivo biodistribution of NPs (Fig. 5j–l and Supplementary Fig. 13). After the intravenous administration of chiral NPs into Balb/c mice, the amounts of L-NPs and D-NPs in the blood gradually decrease over time due to clearance (Fig. 5j). Interestingly, L-NPs exhibited a longer circulation time than D-NPs within 90 min. Moreover, surface chirality also influenced tissue biodistribution and clearance significantly. Most administrated NPs will end up in the liver[47]. Both two chiral NPs reached maximum accumulation in the liver at 30 min due to similarities in SC composition, particularly for albumin. However, the D-NPs can be rapidly and efficiently removed from the liver for 360 min, while L-NPs remained in the liver and a long time was necessary for the removal by the liver. The different levels of Cu in the liver can be explained by differences in circulation time for two chiral NPs (Fig. 5k). Similar tendencies were observed for chiral NP accumulation and removal in the spleen (Fig. 5l). Overall, NP chirality largely influences tissue accumulation and clearance rate in the body.

We further studied the correlation between cellular uptake of chiral NPs and in vivo distribution. In mouse serum-supplemented media, chiral NPs were incubated with RAW 274.7 macrophages for 30 min to study cellular uptake (Supplementary Fig. 14a). To verify the chirality-specific cellular uptake, we determined that more D-NPs were uptaken (30 min) (Supplementary Fig. 14b). This result is consistent with in vivo biodistribution results (Fig. 5j–l), demonstrating that the role of chirality in mediating recognition by macrophages in vivo.

The formation and evolution of chirality-specific HC and the subsequent SC may correlate with in vivo outcomes. To better understand how soft corona on chiral NPs mediates their biodistribution in vivo, we analyzed the results of time- and chirality-dependent of SC formation. It was well established that SC can form at a very early stage of protein corona formation[48]. At an early time point, two chiral NPs showed quite a distinct abundance in SC and HC proteins (Figs. 3c, d and 4c, d), which might address why D-NPs have a faster clearance in the blood even at 3 min (Fig. 5j). We assume that acute phase proteins in the SC on D-NPs at 3 min may be first recognized and then quickly cleared from the blood. By further enriching immunoglobulins in the SC, the accumulation of D-NPs in the liver and spleen was much faster than L-NPs. Similarly, the enrichment of acute phase protein in the SC on L-NP at 30 min can explain its maximum accumulation in the liver. We concluded that the variable level of opsonins and dysopsonins on the chiral NPs regulates the biological fate of these NPs. Surface chirality is one important property of NMs, however, little is known about whether surface chirality mediates the

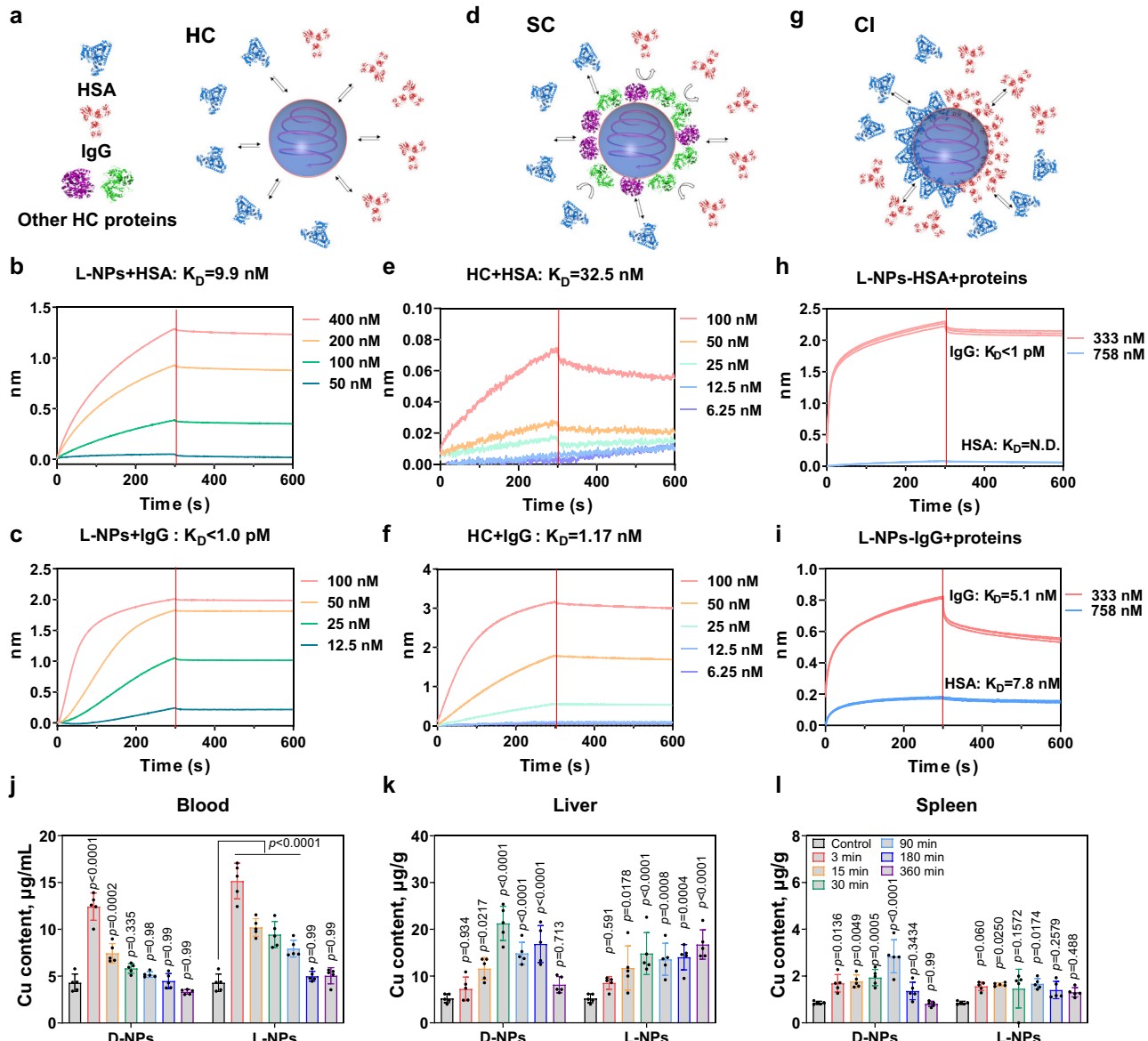

**Fig. 5 | Binding affinity of HC and SC proteins, and tissue biodistribution of the chiral NPs. a–c** Schematic illustration for hard corona formation (**a**) and binding affinity of HSA (**b**) and IgG (**c**) at various concentrations (12.5–400 nM) to L-NPs. **d–f** Schematic illustration for soft corona formation (**d**) and binding affinity of HSA (**e**), and IgG (**f**) at various concentrations (6.25–100 nM) to hard corona–NPs complex. In the presence of 10% mouse serum, proteins are pre-adsorbed on L-NPs at 3 min incubation and then are partly rinsed by water that removes unstable proteins at the outer layer and exposes the hard corona. **g–i** Schematic illustration of competitive interactions (CI) between HSA and IgG on the surface of L-NPs (**g**) and binding affinity of HSA (**h**) and IgG (**i**) to HSA- and IgG-preincubated L-NPs. HSA and IgG are the proteins with relatively high abundance both in the 10% serum and

on the NPs' corona. The spherical structures with gray blue refer to $Cu_2S$ NPs; the structures with blue and red colors represent HSA and IgG; while the structures with purple and green colors represent other HC proteins. **j–l** Time-dependent trafficking process and tissue distribution of chiral NPs after intravenous administration. After tail vein injection of both NPs, Balb/c mice (*n* = 5) are sacrificed to collect blood and organs at 3–360 min. Cu content in the blood (**j**), liver (**k**) and spleen (**l**) is determined by ICP-MS. Statistical significance is calculated by two-way ANOVA with Tukey's multiple comparisons test. *$p < 0.05$; **$p < 0.01$; ***$p < 0.001$; ****$p < 0.0001$; n.s., not significant ($p > 0.05$). All data are shown as mean values and standard deviations for five biological replicates (*n* = 5). Source data are provided as a Source Data file.

trafficking, metabolism, and clearance of NMs in vivo. In this work, the BLI-based Fishing method reveals chirality-characteristic HC and SC components and dynamic change with time. Two chiral NPs showed distinct enrichments of immunoglobins in SC which can well explain why D-NPs have a fast blood clearance rate. In conclusion, surface chirality mediates the formation and evolution of specific SC that determines the ADME of NMs and shows the important role of SC in regulating the fates of NMs and biological effects.

We believe that the collective coronal fingerprint at the nano–bio interface determines the distribution, clearance, and coronal remodeling in the blood. Similarly, the enrichment of acute phase and

immunoglobulins on NMs shortened blood circulation time and enhanced accumulations in organs, such as the liver and kidney[49]. We assumed that the surface properties of NMs are essential to regulating the dynamic evolution of HC at an early stage, which further determines its SC composition. Therefore, these dynamic coronal exchanges are critical for in vivo recognitions. Interestingly, we observed that protein corona formation and remodeling is a stereoselective process, that enhances over time, however, further studies are required. It should be noted that coronal evolution might depend on physiological fluid conditions, size, shape, surface properties, and surface coatings of NPs. The relationship of physicochemical properties with HC and SC

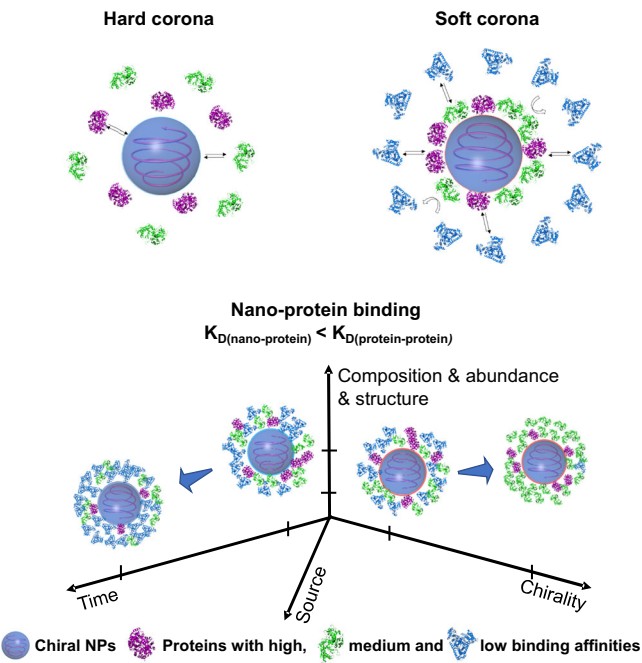

**Fig. 6 | Schematic illustration of time- and surface property-based evolution of protein corona.** Protein corona formation and evolution is a spatiotemporal process, in which the outer and inner layers organize the structure of protein corona companying with dynamic exchange of coronal composition and protein abundance with time, surface properties, source, etc. The interaction between nanoparticles and proteins and between NP-proteins and proteins drives the formation of hard and soft coronas, for which both interaction systems exhibit distinct binding affinity. The formation of the soft and hard corona is a time- and surface coating-dependent thermodynamic and kinetics process, in which the components and the number of protein species of both corona change with time, and the surface properties of nanoparticles such as surface chirality are also a dominant factor for the evolution process. In addition, the source of biological fluids, protein abundance, and other factors are involved in corona formation and evolution. The spherical structures with gray blue refer to $Cu_2S$ NPs; while the structures with purple, green, and light blue, represent various proteins with high, medium, and low binding affinity to NPs.

components will help predict the biological fates of NMs and responses that paves a way for the rational design and application of nanomedicine.

Until now, dynamic information about the formation of hard and soft coronas (biological identity) and their relationship with synthetic or physicochemical identity has been missing. Herein, we report a BLI-based Fishing method for real-time monitoring of protein corona formation and rapid separation of SC and HC components in situ. By coupling this strategy with LC–MS/MS, we successfully identified multilayers of coronal proteins at the nano-protein or protein–protein interface at different time points. Surprisingly, the identified proteins could be found in several corona layers simultaneously, suggesting variable binding affinities with NPs or neighboring proteins. In fact, the NP surface directly mediates HC formation due to the higher binding affinity of nano-protein components, while further SC formation is driven by the lower binding affinity between HC components and various proteins. Moreover, the heterogeneity of HC tended to decrease over time. Interestingly, NP surface chirality mediated the formation of specific SC and HC characteristics in a time-dependent manner. We believe that collective re-modeling and dynamic evolution of coronal composition, i.e., dysopsonins and opsonins, at the nano–bio interface is the key to long blood circulation or for faster clearance in vivo. These findings highlight the importance of this Fishing strategy in supporting the exploration of rapid and continuous adsorption, exchange, and rearrangement of corona composition and displaying promising

applications for the study of the dynamic evolution of protein coronas and surface-specific corona formation (Fig. 6).

Importantly, the Fishing strategy can be extended to a simple and universal method to study how physicochemical properties of NPs (including elemental composition, surface charges, modification, surface roughness, size, etc.) affect the formation of HC/SC and biological responses. For example, the BLI-based Fishing method can be used to in situ characterize corona formation on specific nanomaterials such as quantum dots, ultra-small NPs, ultrathin 2D nanosheets, soft nanomaterials with low density that are difficult to be centrifugated to isolate NPs-corona from the protein solution. Moreover, this approach will offer opportunities to study different coronas for nanomaterials in various biological fluids/microenvironments (plasma, cerebrospinal fluid, pulmonary fluid, gastrointestinal fluid, etc.). In future works, the dynamic exchange of protein coronas in different biological sources could be also investigated by this strategy that helps deeply understand corona evolution during transfer among different fluids and is powerful for the study of biological effects of nanomaterials and the developments of nanomedicines.

## Methods
### Materials
Copric chloride dihydrate (ACS), L-/D-cysteine hydrochloride monohydrate (98%), and thioacetamide (98%) were purchased from Aladdin Biochemical Technology Co., Ltd. (Shanghai, China). Trifluoroacetic acid (≥99.9%, HPLC Grade, TFA), isopropyl alcohol (≥99.5%), and β-Mercaptoethanol (99%) were purchased from Macklin Biochemical Co., Ltd. (Shanghai, China). Immunoglobulin G, Albumin, and Transferrin purified from human serum were purchased from Sigma-Aldrich (Darmstadt, Germany). Ethanolamine (EA, ≥ 99.0% ACS), 1-ethyl-3-(3-dimethylaminopropyl) carbodiimide hydrochloride (EDC, ≥ 98.0%), and N-hydroxysuccinimide (NHS, ≥ 98.0%) were purchased from Sigma-Aldrich.

### Synthesis of chiral cysteine-capped $Cu_2S$ NPs
Chiral NPs were synthesized according to the following protocol. In general, 1 mmol of $CuCl_2·2H_2O$ and 2 mmol of L-/D-cysteine hydrochloride monohydrate was added to 100 mL of Milli-Q water in a round-bottomed flask and dissolved by mixing at 30 °C for ~30 min. Ten mL of the reaction solution was then added to 90 mL of Milli-Q water and mixed at 50 °C for ~10 min under an $N_2$ atmosphere, after which the pH value was adjusted to ~8.0 using 1 M NaOH until the color changed to clear yellow or brownish. Finally, 0.05 mmol of thioacetamide aqueous solution was added to the reaction with vigorous stirring under constant $N_2$ for 30–40 min to generate the final products. After cooling to room temperature (RT), light-orange chiral NPs were isolated by mixing the sample with isopropanol at a ratio of 1:1 (v/v) and spinning at 9600×g for 5 min. The precipitate was collected and suspended in water until further use.

### Characterization of physicochemical properties of chiral NPs
Inductively coupled plasma−mass spectrometry (ICP−MS) was used to quantify the concentration of Cu in the NPs. In detail, samples were mixed with 70% nitric acid (5 mL) overnight and boiled at 150 °C for 3 h with the addition of 2–4 mL of $H_2O_2$ to each sample. Finally, samples were treated with a mixed acid solution containing 2% (v/v) $HNO_3$ and 1% (v/v) HCl at a final volume of 3 mL, which were measured in triplicate using a Thermo Elemental X7 ICP-MS.

To ensure morphology and size distribution, a transmission electron microscope (TEM), named FEI Talos F200S and operated with an accelerating voltage of 200 kV, was used for high-resolution imaging measurements. TEM grids were prepared by drop-casting 10 μL of chiral NP dispersion in a water−methanol mixture (25-75% v/v) at a final concentration of 0.25 mg/mL and drying overnight prior to TEM analysis. In different fields of view, NPs (n = 100) were randomly observed

by TEM. The diameter of NPs was measured and recorded according to Image J (version 1.53k, National Institutes of Health, USA), which was used to calculate the average size of NPs.

To characterize the chirality, circular dichroism (CD) spectra were obtained with a J-1500 CD spectrometer (Jasco, Japan) under high-purity nitrogen at 25 °C for all measurements, using a scanning range of 195–500 nm.

To characterize the electronic structures, X-ray photoelectron spectroscopy (XPS) measurement was performed using a Thermo Scientific ESCALAB 250e III. Before XPS analysis, the powder of NPs was mounted and adhered to an aluminum foil to achieve a uniform surface for analysis. The survey spectra represented an average of 10 scans taken with a pass energy of 200.00 eV and a step size of 1 eV. Both survey scans were performed to assess Cu and S elements in detail. In addition, to obtain the crystal structure, X-ray diffraction (XRD) patterns were recorded on STOE STADI MP diffractometer in transmission mode with Mo Kα radiation ($\lambda = 0.07093$ nm).

### Incubation of chiral NPs with serum

The female BALB/c mice ($n = 5$, 6–8 weeks) blood was collected from retro-orbital sinus and was clotted at room temperature without any agitation for 20–30 min (for details see below). Blood was centrifuged at $1100 \times g$ for 10 min at 4 °C to obtain serum, then serum was purified by centrifugation at $14,000 \times g$ for 5 min to remove any insoluble debris and aggregated proteins. 0.5 ml of serum aliquots were stored at −80 °C before further experiments. Solutions of 10% mouse serum and NPs (~0.5 mg, $n = 3$, 100 μL) were incubated at 37 °C for 3 and 30 min to make comparisons between Fishing- and centrifugation-based coronal isolation approaches. In detail, the centrifugation was performed to isolate coronal–NPs complexes from free or loosely bound proteins ($360,000 \times g$, 1 h at 4 °C). Pellets were then washed three times with ice-cold PBS, and eluted with washing buffers for further analysis.

### Eluting HCC proteins from NP–corona complex

Washing buffer (WB) containing 5% SDS (100 μL) was added to NP–corona complex to recover proteins from chiral NPs. Samples were heated at 95 °C for 15 min to denature and strip proteins from chiral NPs. Then, the samples were centrifuged at $11,700 \times g$ for 15 min. HCC proteins (~20 μg) were further eluted and digested for label-free LC–MS/MS analysis.

### Protein corona collection with BLI-based Fishing method

BLI measurements were obtained with an Octet RED96e system (ForteBio, USA). AR2G biosensors from ForteBio were pre-wetted for 10 min, after which human IgG was coated on the biosensors using the amino-coupling method, and chiral NPs were loaded. In brief, (1) Pre-wet sensors (10 min) were immersed for 60 s in buffer alone. (2) Sensors were activated for 300 s with a 1:1 (v/v) ratio of EDC/NHS. (3) Protein Loading: sensors were immersed for 300 s in IgG-containing buffer (pH 5.5, 50 μg/mL). (4) Quenching: sensors were immersed for 300 s in EA buffer solution (pH 8.5, 1 M). (5) Baseline: sensors were immersed for 60 s in buffer alone. (6) L-/D-NPs loading: sensors were immersed in the suspension of chiral NPs for 300 s, and then chiral NPs were loaded. (7) Baseline: sensors were immersed in the buffer for 60 s alone. Chiral NP-loaded sensors were further introduced to 10% mouse serum for corona formation (3 or 30 min), followed by protein separation in washing buffer (WB, 20 s). To exclude nonspecific protein adsorption to the sensor itself, 10% mouse serum was dissolved in 0.05% tween-20 buffer. Alternatively, to exclude nonspecific adsorption of proteins to the IgG-immobilized sensors, the sensors can be immersed in 0.05% tween-20 buffer for 30 s, loaded with chiral NPs, then introduced to 10% mouse serum for corona formation (3 or 30 min). Aqueous TFA solutions (0.005–0.1%) were used as WB for this project. SC and HC were collected via two-step isolations with 0.005% and 0.1% WB, respectively. To fully explore the corona formation

process, different layers of the corona, (namely, soft and hard coronas) were studied by the BLI-based Fishing method. To better mimic the physiological environment, the Fishing experiments were run under a continuous shaking of the sample plate (1000 rpm) at 37 °C that ensure the homogenous concentration of serum proteins in the sample solution during corona formation.

### LC–MS/MS analysis

For LC–MS/MS analysis, digested products were separated with a 120 min gradient elution at a flow rate of 0.300 μL/min using a Thermo Ultimate 3000 nano-UPLC system directly interfaced with a Thermo Fusion LUMOS mass spectrometer. The analytical column was an Acclaim PepMap RSLC column (75 μm ID, 250 mm length, C18). Mobile phase A consisted of 0.1% formic acid, and mobile phase B consisted of 80% acetonitrile with 0.1% formic acid. The Fusion LUMOS mass spectrometer was operated in data-dependent acquisition mode using Xcalibur 4.1.50 software, and there was a single full-scan mass spectrum in the Orbitrap (350–1800 $m/z$, 60,000 resolution), followed by 20 data-dependent MS/MS scans. MS/MS spectra from each LC–MS/MS run were searched against the selected database using the software Proteome Discoverer (Thermo Scientific, version 2.4). Positive ion spray voltage was set to 2200 V, with the ion transfer tube temperature at 320 °C. The detector was set to Orbitrap with a mass range of 350–1800 $m/z$, RF lens 40%, and AGC target at 3E6. In this study, trypsin was used, thus the cleavage sites are K and R. The maximum number of missed cleavage sites was set to 2. Oxidation and Carbamidomethyl modifications were used. Mass tolerance for precursor and fragment ions was set as 10 ppm for the precursor and 0.02 Da for the fragment. The minimum number of peptides with a length of 6–144 amino acids for protein identification was set as one. Identified corona proteins on chiral NPs were manually assigned via searching against the mouse protein database in the UniProt database downloaded from UniProt. The score cutoff was set to automatic mode. Peptide FDR was set as 0.01 and protein FDR was set as 0.05.

In this study, triplicate biological replicates ($n = 3$) of 13 samples and 4 controls (i.e., bare and IgG-coated biosensors at 3 and 30 min) were analyzed by LC–MS/MS. Total proteins (2 μg) in each tested sample were normalized by quantification prior to the digestion of proteins. On both control sensors, a neglectable amount of albumin, one of the abundant proteins in mouse serum, was detected because the control samples (containing 1.2–1.4% albumin in total adsorbed proteins) should be concentrated ~20 times prior to analysis. Complement C1s protein (92.2–94.8% in total adsorbed proteins) was considered as a non-specific binding protein of the control sensors upon interaction with 10% mouse serum. The proteomics data of the bare biosensor itself and IgG-coated biosensors are presented in Supplementary Data 1 and ProteomeXchange Consortium (see the "Data availability" section). The percent abundance for each protein was calculated based on label-free quantification intensities relative to the total sum of protein intensities for each sample.

### Protein binding affinity on chiral NPs

AR2G biosensors were immobilized with human IgG (50 μg/mL). All proteins were introduced to the L-NPs at different concentrations (12.5–400 nM). Kinetic analysis using L-NPs loaded AR2G biosensors was performed as follows: (1) Baseline: sensors immersed for 60 s in buffer alone. (2) Association: 300 s immersion in a solution containing proteins at different concentrations. (3) Dissociation: 600 s immersion in the buffer. Curve-fitting was performed using a 1:1 binding model with data analysis software (ForteBio version 11.0). Mean $K_D$ values were determined by fitting all binding curves with an $R^2$ value ≥0.99.

### Binding affinity of SCs on L-NPs

We used the Octet Red96e instrument (ForteBio, USA) at 30 °C with shaking at 1000 rpm. AR2G biosensors were coated with human IgG

according to the manufacturer's protocol before further loading of chiral L-NPs (as described above). Curve-fitting was performed using Octet Data analysis software (ForteBio version 11.0). Affinity measurement data for competitive binding was presented as an average of three independent samples. Kinetic analysis using the L-NPs loaded AR2G biosensor was performed as follows: (1) Baseline: 60 s immersion in the buffer. (2) Loading: 300 s immersion in 10% mouse serum solution. (3) Association: 300 s immersion in solution with 6.25–100 nM HSA or IgG proteins. (6) Dissociation: 600 s immersion in the buffer. Curve fitting was performed using a 1:1 binding model using data analysis software (ForteBio). Mean $K_D$ values were determined by fitting all binding curves with an $R^2$ value ≥0.99.

### Competitive interactions among different proteins
L-NPs were incubated with HSA (758 nM) or IgG (333 nM) for 3 min to form an initial HC layer. To determine protein–protein binding affinities, the pre-incubated L-NPs were incubated with those proteins once more at constant concentrations (for details refer to the section above).

### Protein graphics
Proteins were generated, and color modified in PyMOL Molecular Graphics System (version 2.2) based on PDB files (PDB ID: 1SI4[50], 1AO6[51], 1IGY[52], 3GHG[53], and 1D3K[54]) were obtained from the RCSB Protein Data Bank (PDB, www.rcsb.org) and are used in Figs. 1i, j, 3a, 4a, 5a, 6, and Supplementary Figs. 2 and 3 to illustrate protein corona formation or isolation. Of note, random colors were chosen for the proteins to visualize various proteins with different binding affinities.

### Cell culture medium
Penicillin–streptomycin (PS), phosphate buffer saline (PBS), and DMEM cell culture medium were purchased from Wisent Corporation (Wisent, Canada). Fetal bovine serum (FBS) was purchased from Biological Industries (BI, Israel).

### Cell culture
RAW 264.7 mouse monocyte/macrophage-like cells (catalog number SCSP-5036) were obtained from the National Collection of Authenticated Cell Cultures (Shanghai, China). RAW 264.7 cells were cultured in DMEM medium supplemented with 10% FBS and 1% PS in a humidified 5% $CO_2$ atmosphere at 37 °C.

### Cellular uptake of D-NPs and L-NPs
RAW 264.7 macrophages were seeded and cultured on six-well plates at a density of 400,000 cells/well overnight at 37 °C and 5.0% $CO_2$ prior to the uptake study. For these studies, a complete cell culture medium containing 10% mouse serum was used. Cellular uptake of chiral NPs was further evaluated using ICP–MS, by which the experimental details were indicated below.

### Tissue distribution and accumulation of chiral NPs
Pathogen-free BALB/c mice (female, 6–8 weeks old) were purchased from Beijing Vital River Laboratory Animal Technology. All the animals were maintained on a standard diet and water ad libitum at 22 ± 2 °C and 50–60% relative humidity on a 12 h light/12 h dark cycle. All protocols were approved by the Institutional Animal Care and Use Committee of the National Center for Nanoscience and Technology and performed under the ethical guidelines for the use and care of animals. Female BALB/C mice (n = 5, 6–8 weeks) were used in in vivo biodistribution and proteomics studies. In detail, female Balb/c mice (n = 5, 6–8 weeks) were intravenously injected with L-NPs or D-NPs at a dose of 0.35 mg/kg, which was normalized by the Cu concentration. Blood and organs (heart, lung, liver, kidney, and spleen) were harvested 3-, 15-, 30-, 90-, 180- and 360-min post-injection and stored at −80 °C prior to solution preparation and ICP–MS measurements.

Accumulation of NPs in organs was further evaluated based on Cu concentration using ICP–MS[7]. In brief, the weighted samples of organs and blood, and the counted cells were mixed with nitric acid (70%, 5–7 mL) overnight in a triangular flask. The samples were then heated on a heating plate at 150 °C for 3 h, during which 4 mL of $H_2O_2$ was gradually added and the mixture solution was evaporated to a volume <0.5 mL. Finally, a mixed aqueous solution containing 2% (v/v) $HNO_3$ and 1% (v/v) HCl was added to the above mixture to a volume of 3 mL before ICP–MS measurement. Each sample was measured in triplicate using a Thermo Elemental X7 ICP–MS.

### Statistical analysis
To assess the statistical significance of differences, ANOVA analysis was performed using GraphPad Prism (version 8.2.1, USA), and the $p < 0.05$ was considered statistically significant. The statistical tests used to conduct each analysis in the study are described in the corresponding figure legends. If significance was determined, a two-way ANOVA post-hoc multiple comparison analysis was conducted with the Tukey test.

### Reporting summary
Further information on research design is available in the Nature Research Reporting Summary linked to this article.

## Data availability
The data generated in this study is available with the article, Supplementary information, and Supplementary data. All other data are available from the corresponding authors upon request. The mouse proteomic dataset is available on the UniProt website (https://www.uniprot.org). The proteomic data generated by mass spectrometry, including raw data and search results have been deposited to the ProteomeXchange Consortium (http://proteomecentral.proteomexchange.org) via the PRIDE partner repository with the dataset identifier: PXD033976[55]. PDB files 1SI4, 1AO6, 1IGY, 3GHG, and 1D3K were obtained from the Protein Data Bank (PDB). The images of tubes (Fig. 2a, Supplementary Fig. 13a), mouse and syringe (Supplementary Fig. 13a), three neck flasks (Supplementary Fig. 1a) were adopted from SMART-Servier Medical Art (https://smart.servier.com/). The identified coronal protein parameters, including molecular weight, isoelectric points, and relative abundance (in Supplementary Figs. 4, 8, 9 and Figs. 2b, 3b, 4b) generated in this study are presented in Supplementary Data 1. Data processing and cluster analysis are performed using R language (version 4.0.5). Protein abundance is normalized to reduce the effect of outliers and minimize overall proteome differences. Hierarchical clustering analysis is performed by the R package heatmap on the normalized, log2 abundance using Euclidean distance. The source data including Figs. 1e–h, 2b, 3b, 4b, 5b, c, e, f, h, i, j–l, and Supplementary Figs. 1b–d, 2, 3, 12, 13b–d, 14b are provided in a Source Data file. Source data are provided with this paper.

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

## Acknowledgements

We appreciated Bai Li (IHEP) for his assistance with ICP–MS measurement, Dongqi Ni (NCNST) for his help with animal experiments, and Liuxiang Wang (NCNST) for his help with heatmap analysis. We appreciate funding from the National Key R&D Program of China (grant no. 2021YFA1200900 to C.C. and L.W.; grant no. 2020YFA0710702 to L.W.), the National Natural Science Foundation of China (grant no. 31971322 to L.W., grant no. 22077008 to J.W. and grant no. 21773172 to Y. Zhou), Start Funding of Wenzhou Institute of UCAS (grant no. WIU-CASQD2019001 to Y. Zhou), Major Instrument Project of National Natural Science Foundation of China (grant no. 22027810 to C.C.), Beijing Municipal Health Commission (grant no. 2021-1G-1191 to L.W.), CAS President's International Fellowship Initiative (PIFI, grant no. 2021PM0059 to L.W.), Science and Technology Innovation Project in IHEP (grant no. E25459U210 to L.W.), NSFC Major Research Plan-Integrated Program (grant no. 92143301 to C.C.), the CAMS Innovation Fund for Medical Sciences (grant no. CIFMS 2019-I2M-5-018 to C.C.), the Strategic Priority Research Program of Chinese Academy of Sciences (grant no. XDB36000000 to C.C.), and the Research and Development Project in Key Areas of Guangzhou (grant no. 202008070007 to C.C.), the Open Fund from State Key Laboratory of Natural and Biomimetic Drugs in Peking University (grant no. K202001 to L.W. and J.W.), Guangdong high level Innovation Research Institute (grant no. 2020B0909010001 to C.C.), and the Science and Technology Innovation Project for Undergraduate Students in IHEP and UCAS (grant no. H95120P0U7 to L.W.). This work was partially supported by the Directional Institutionalized Scientific Research Platform relies on the Beijing Synchrotron Radiation Facility of the Chinese Academy of Sciences.

## Author contributions

This project was conceived and designed by D.B., J.W., L.W., and C.C., and was supervised by C.C., L.W., Y. Zhou, and Y. Zhao. Main experiments were performed by D.B. and J.W. with assistance from J.Z., Y.C., Y.L. LC–MS/MS experiments were performed and analyzed by D.B., X.S., and X.Z. ICP–MS experiments were performed and analyzed by D.B., K.L., X.L., and R.Q. Manuscript was prepared by D.B., J.W., L.W., Y. Zhou, and C.C. with input from other authors. All authors have proofread the paper and approved its publication in the present form.

## Competing interests

The authors declare no competing interests.
