## [Peer Review File · Nature Communications]

REVIEWER COMMENTS

Reviewer #1 (Remarks to the Author):

This paper is focused on the development of a platform, consisting chiral Cu₂S nanoparticles that were loaded onto a biosensor, for in situ monitoring of soft corona. Robust identification of soft corona proteins on the surface of nanoparticles is of crucial importance in prediction of the safety and diagnostic/therapeutic capacity of nanosystems. Although the core concept of the paper is promising, the presented data/claims lack robustness due to the following reasons:

1- Some essential controls are missing; for example, the proteomics data on the formation of corona on the biosensor, itself, is missing. There should be two important controls here: i) the sensor, itself, goes through incubation with serum and the washing steps; and ii) the IgG coated biosensor (not loaded with nanoparticles) goes through incubation with serum and the washing steps. I suspect that the observed contribution of albumin and its significant changes over incubation time in mass-spect outcomes may not come from nanoparticles' protein corona.

2- Although the system is innovative, it doesn't represent the actual environment for protein corona formation. In a normal protein corona formation environment, nanoparticles are mixed with the biological fluids in a dynamic situation (e.g., vial agitation). Nanoparticles that are loaded on the surface of the biosensor cannot move within the biological fluids. This means that after first interactions with proteins, there would be a protein concentration gradient in the vicinity of biosensor' surface which controls the further protein-nanoparticle interactions.

3- In order to correlate the protein corona outcomes with nanoparticles' in vivo outcomes, the actual plasma/serum (e.g., in this case, mice) should be used for corona analysis. It is well-documented in the literature that the composition and decoration of protein corona is highly dependent to the type of biological fluids. In other words, the protein corona data from FBS cannot be used to predict biological fate of nanoparticles in mice.

Some other minor comments:

1- Some critical information regarding the formation of protein corona (e.g., the temperature) in the biosensor platform is missing.

2- Lack of consistency in the units (e.g., centrifugation speed, e.g., the authors used both "g" and "rpm" with no clear reason for this interchange).

Reviewer #2 (Remarks to the Author):

The authors presented an original work in which they develop a fishing method for monitoring the dynamic evolution of the protein corona, with a special focus on the role of the soft corona. I appreciated the work of the authors. However, there are major concerns regarding the experimental part.

My specific comments to the authors:

1) To isolate the soft corona, 0.005% washing buffer is used. It is not clear how total corona is isolated differently from the hard corona, due to that for both TC and HC 0.1% washing buffer is used.

2) Hard corona. When using the fishing method, the authors perform an incubation of NPs in 10% FBS at 2-time points (3 minutes and 30 minutes), identifying by LCMSMS 262 (3 min) and 217 (30 minutes) proteins for D-NPS and 255 (3 min) and 197 (30 minutes) for L-NPs. When using the centrifugation method (to make comparisons between the 2 approaches), the authors perform an incubation in 10% serum using only one-time point (1h at 37 degrees), different from those used in the fishing approach. This is not appropriate. The same exact conditions must be used when comparing the two approaches. In addition, what does 10% serum mean? Is it 10% FBS? If so, please use the same language. Moreover, at which temperature the Fishing experiment has been performed? Temperature is another important parameter and must be the same. The results of the centrifugation experiments for HC formation performed in the same conditions of the fishing approach are required.

3) Figure 2 and 3. The pie charts should have the same color legends (e.g. albumin pink/grey)

4) Soft corona. When using the fishing method for soft corona formation, the authors use 2-time points (3 min and 30 min) identifying by LCMSMS 394 (3 min) and 288 (30 minutes) proteins for D-NPS and 424 (3 min) and 324 (30 minutes) for L-NPs. As expected, most of the identified proteins (90%) are common to those identified in the HC proteins, thus demonstrating that the different concentration of washing buffer is not a valid method to separately elute soft and hard corona.

The authors should consider revising the entire work from another angle: using the time (3 minutes vs 30 minutes) as a factor for discriminating soft and hard corona. Indeed, it is well established that the soft corona is formed in the very early stage of PC formation and that later this is replaced by the hard corona. (see Tenzer S. et al Nat Nano 2013)

5) Despite much information could be obtained from this work, I don't see a great novelty in it. The authors should emphasize the novelty of such work if they aim to publish it in a high IF journal.

Reviewer #3 (Remarks to the Author):

The outer layer of protein coronas, i. e. soft corona contributes to “biological” identity of NPs that regulates fundamental effects and fates of NPs in vivo and in vitro, however, little is known about the soft corona composition due to a longstanding challenge in soft corona analysis. This work made a breakthrough in soft corona analysis, in which authors presented a simple method to precisely capture several layers of proteins on NPs in complex biological matrix or media. Combining BLI-biosensor with MS, the strategy can monitor the formation and exchange of protein corona in real time that helps fast isolation of proteins at outer and inner layers and MS analysis to identify protein composition. This study is interesting and has potential broad applications in multidisciplinary fields such as nanobiology, nanomedicine, biophysics, and environment science, etc. I would like to recommend its publication after minor revision.

1) The “Fishing” strategy is useful in studying the relationship between surface properties of NPs and biological responses due to corona formation. Besides chirality, authors may point out more applications in the discussion to explain the importance of the method.

2) About the novelty of method, it is better to compare this method more with other methods in discussion or in the introduction.

3) Check and write down proper abbreviation and full names such as BLI.

4) Chiral molecules are close to the adsorbed HC proteins and may affect the formation of HC. Authors can discuss more about chirality-based HC protein “fingerprint” about the type subtypes.

5) They used wash buffer to elute the proteins at outer layer and inner layer. They may show BLI data about the adsorption and desorption of NPs on the sensors to show the stability of NPs during the wash step.

Responses to the Reviewers' Comments

Reviewer #1

This paper is focused on the development of a platform, consisting chiral Cu₂S nanoparticles that were loaded onto a biosensor, for *in situ* monitoring of soft corona. Robust identification of soft corona proteins on the surface of nanoparticles is of crucial importance in prediction of the safety and diagnostic/therapeutic capacity of nanosystems. Although the core concept of the paper is promising, the presented data/claims lack robustness due to the following reasons:

1. Some essential controls are missing; for example, the proteomics data on the formation of corona on the biosensor, itself, is missing. There should be two important controls here: i) the sensor, itself, goes through incubation with serum and the washing steps; and ii) the IgG-coated biosensor (not loaded with nanoparticles) goes through incubation with serum and the washing steps. I suspect that the observed contribution of albumin and its significant changes over incubation time in massspect outcomes may not come from nanoparticles' protein corona.

Response. We appreciate the reviewer's positive comments. As suggested, we have prepared two control groups for proteomics analysis including the bare biosensor itself (i) and IgG-coated biosensor without nanoparticles (ii) and presented the updated results in this revision. After incubation with mouse serum and washing steps, the total amount and concentration of isolated proteins from the bare (i) and IgG-coated (ii) sensors were much less compared to NPs-sensors. **The amount of adsorbed proteins on the bare (i) sensors is only 5% and 3.7% of that on NPs at 3 min and 30 min; while the amount of adsorbed proteins on the IgG-coated (ii) sensors is only 5% and 5.6% of that on NPs at 3 min and 30 min.** Thus, the albumin content in corona formation on the biosensor can be negligible and can well exclude the above speculation "the contribution of albumin and its significant changes over incubation time in mass spectrum outcomes may not come from nanoparticles' protein corona".

Here, we describe the details for your consideration. (1) We firstly incubated the AR2G biosensor itself and IgG-coated biosensors with 10% mouse serum for 3 and 30 mins. The adsorbed proteins were desorbed in the presence of 200 μ l washing buffer

(WB). The amount of adsorbed proteins from the NPs, i.e., the total amount of proteins separated from NPs per sensor is $\sim 4 \mu\text{g}$ and $\sim 5.4 \mu\text{g}$ at 3 and 30 min, respectively. (2) Comparably, the total amount of proteins separated from either bare (i) or IgG-coated (ii) sensors is too low to be measured by Nanodrop. (3) To obtain enough amount of proteins from the control sensors (i, ii), we collected a total volume of 4 ml protein solutions from 20 sensors (200 μl each experiment) to satisfy the Nanodrop measurement. The 4 ml of protein solutions were further concentrated to 200 μl by ultrafiltration and the concentration was quantified according to Nanodrop. (4) For the bare sensor (i), the actual amount of adsorbed proteins per sensor was calculated to be $\sim 0.2 \mu\text{g}$ at 3 and 30 min, respectively. For IgG-coated (ii) sensors, the actual amount of adsorbed proteins per sensor was $\sim 0.2 \mu\text{g}$ and $\sim 0.3 \mu\text{g}$ at 3 and 30 min, respectively. (5) Thus, the amount of adsorbed proteins on the bare sensor (i) is only 5% and 3.7% of that on NPs at 3 min and 30 min incubation, respectively. Meanwhile, the amount of adsorbed proteins on IgG-coated sensors (ii) is only 5% and 5.6% of that on NPs at 3 min and 30 min incubation, respectively.

After proteomics analysis, the identified proteins and their relative abundance of two control sensors in 10% mouse serum system are provided in *Supplementary Data* file. We found albumin makes quite low contribution (1.17-1.46%) to the components of total protein corona on both control sensors after 3 and 30 min incubation. **Thus, we think that the nonspecific adsorption of albumin on the bare biosensor itself (i) and IgG-coated biosensor (ii) can be neglected.** However, interestingly, the significant adsorption of component C1s protein (92.2-94.8% in total adsorbed protein corona) to both control sensors (i, ii) was observed upon interaction with 10% mouse serum. Herein, complement C1s can be considered as a nonspecific protein binding to the sensor on the mouse serum samples.

In addition, we have modified the content in the revised manuscript on Page 22. “Total proteins (2 μg) in each tested sample were normalized by quantification prior to the digestion of proteins. The bare AR2G biosensors and IgG-coated sensors were used as control sensors. On both control sensors, neglectable amount of albumin, one of the abundant proteins in mouse serum, was detected because the control samples (containing 1.2-1.4% albumin in total adsorbed proteins) should be concentrated ~ 20

times prior to analysis. Complement C1s protein (92.2-94.8% in total adsorbed proteins) was considered as a non-specific binding protein of the control sensors upon interaction with 10% mouse serum. The proteomics data of the bare biosensor itself and IgG-coated biosensors are presented in Supplementary Data file and ProteomeXchange Consortium (see the section of Data availability).”

2. Although the system is innovative, it doesn't represent the actual environment for protein corona formation. In a normal protein corona formation environment, nanoparticles are mixed with the biological fluids in a dynamic situation (*e.g.*, vial agitation). Nanoparticles that are loaded on the surface of the biosensor cannot move within the biological fluids. This means that after first interactions with proteins, there would be a protein concentration gradient in the vicinity of biosensor' surface which controls the further protein-nanoparticle interactions.

Response. We are grateful to this reviewer's comments about the significance of this study and the suggestion. Yes, it is quite important to keep the actual environment for protein corona formation as possible. Actually, we indeed kept the temperature and the agitation of the sample plate to maintain the component and concentration homogeneity in 10% serum solution that could well mimic actual environment for protein corona formation from the beginning of the study. However, it was pity that we forgot to mention the detailed information about the experimental parameters to study protein-nanoparticle interactions in previous version. In this revision, we have included these parameters such as the temperature and the agitation of the sample plate in the *Materials and Methods* section.

Herein, several important factors such as fluid, and homogeneous concentration of proteins with increasing incubation time and performed serum-NPs interaction experiments under a continuous agitation of sample plate (1,000 × rpm) at 37°C were considered. Under such environment, the fluid around NPs-loaded sensor changes dynamically in the microplate that enables homogenous components of serum protein and the same concentration of proteins with time. This will realize relatively homogenous and stable systems for protein corona formation.

Thus, we have updated the content on Page 21, “To better mimic physiological environment, the “Fishing” experiments were run under a continuous shaking of sample

plate (1,000 × rpm) at 37 °C that ensured the homogenous concentration of serum proteins in sample solution during the corona formation.”

3. In order to correlate the protein corona outcomes with nanoparticles' *in vivo* outcomes, the actual plasma/serum (*e.g.*, in this case, mice) should be used for corona analysis. It is well-documented in the literature that the composition and decoration of protein corona is highly dependent to the type of biological fluids. In other words, the protein corona data from FBS cannot be used to predict biological fate of nanoparticles in mice.

Response. We appreciated this reviewer's constructive suggestion. We totally agreed that the composition and decoration of protein corona is highly dependent to the type of biological fluids. As suggested, we have added new experiments to study the corona in 10% mouse serum instead of 10% FBS that not only demonstrate analytical method for SC and HC *in situ*, but also better correlate protein corona formation to the fates of NPs *in vivo*. Moreover, we have replaced the previous data for protein corona using 10% FBS by new data using 10% mouse serum. These new sets of data include the components of HCC, HC, and SC on the NPs and its evolution at different interaction time.

As below, we listed the updated major results for 10% mouse serum that well explained the fates of NPs *in vivo*. To conclude, with increasing incubation time, more opsonin molecules such as immunoglobulins are found in the SC on D-NPs than those on L-NPs (*Figs. 4C, 4D*). This result can explain why D-NPs show a faster clearance rate in blood than L-NPs (*Fig. 5J*) and faster accumulation in the liver and spleen (reticuloendothelial systems).

The updated data:

- HCC: *Figs. 2B-G, Supplementary Figs. 5-7*
- HC: *Figs. 3B-D, Figs. 5E-F, Supplementary Figs. 7-12*
- SC: *Figs. 4B-D, Supplementary Figs. 8-12*
- Cell uptake of chiral NPs in cell culture medium containing 10% mouse serum: *Supplementary Fig. 14*
- Proteomics data are available in Supplementary Data file and ProteomeXchange Consortium.

4. Some other minor comments:

1) Some critical information regarding the formation of protein corona (*e.g.*, the temperature) in the biosensor platform is missing.

Response. Thanks. We have provided more information about the protein corona formation in the revision.

We updated the content on Page 21, “To better mimic physiological environment, the Fishing experiments were run under a continuous shaking of sample plate (1,000 × rpm) at 37 °C that ensured the homogenous concentration of serum proteins in the sample solution during corona formation.”

2) Lack of consistency in the units (*e.g.*, centrifugation speed, *e.g.*, the authors used both “g” and “rpm” with no clear reason for this interchange).

Response. Thanks. We have confirmed the units during the revision: The centrifugation speed units have been unified to “rpm”.

Reviewer #2

The authors presented an original work in which they develop a fishing method for monitoring the dynamic evolution of the protein corona, with a special focus on the role of the soft corona. I appreciated the work of the authors. However, there are major concerns regarding the experimental part.

My specific comments to the authors:

1. To isolate the soft corona, 0.005% washing buffer is used. It is not clear how total corona is isolated differently from the hard corona, due to that for both TC and HC 0.1% washing buffer is used.

Response. We appreciated this reviewer's comments. In this work, we proposed a novel BLI-based "Fishing" strategy and attempted to isolate soft corona (SC), hard corona (HC), or total corona (TC) from the surface of NPs, i.e., combining a BLI-based NPs loading with gradient washing buffer-based protein elution *in situ*.

Using BLI sensors, we can directly monitor the adsorption and desorption of proteins that will help to optimize the concentrations of washing buffer (WB) and elution steps. WB with gradient concentrations can wash out the adsorbed proteins at the different coronal layers to separate SC, HC, or TC. By comparing the baseline signal after elution with the initial equilibrium signal, we can observe how much the corona can be washed out or the corona remains on the NPs (*Supplementary Fig. 2*).

- TC proteins can be eluted once by 0.1% of WB as a single experiment from the NPs' surface that isolates total proteins (*Supplementary Fig. 2*). In the revision, TC data are not shown because we did not further compare TC with HC, SC, and HCC any more.
- HC proteins should be obtained from the surface of NPs after a two-step elution (*Supplementary Fig. 3*): 0.005% of WB is firstly used to elute the outer layer of corona (SC components), and then 0.1% of WB is used to isolate the rest of corona (HC components).

To clearly show the steps to obtain HC and SC, we have added the description on

Pages 7-8 and updated the description in the revised manuscript as below.

“To clarify corona composition and organization, isolated proteins with weak binding affinity at the outer layer are described as SC, and proteins with stronger binding affinity at the inner layer are described as HC.”

Supplementary Figure 2. Schematic workflow of the isolation step by BLI-based “Fishing” method. Chiral NPs loaded biosensors are firstly equilibrated into buffer solution for 1 min with a further introduction to the biological fluid for 3 or 30 min. Real-time monitoring for the formation (3 min) of the protein coronas on the surface of chiral NPs is shown. Washing buffer (WB) at different concentrations is used for the isolation of the corona proteins from the chiral NPs-loaded biosensors. 0.005% of WB and 0.1% of WB were used to isolate SC and HC, respectively. The shift ($\Delta\lambda$) difference between the baseline and equilibrium steps indicates that chiral NPs remain on the sensor after introducing into the WB. During the adsorption process, the adsorbed proteins increase with time to reach an equilibrium. During the washing step, the proteins can be partly or fully eluted by WB at different concentrations. The NPs for the control (washed by pure water) remain on the sensor, while the signals for NP-protein complex in WB do not decrease at the final baseline compared to the signals at the equilibrium curve, suggesting a stable condition of NPs on the sensor.

Supplementary Figure 3. Schematic workflow of SC and HC isolation by “Fishing” method.

Chiral NPs loaded biosensors are firstly equilibrated into buffer solution for 1 min (not shown) with a further incubation in serum. 0.005% of WB and 0.1% of WB are chosen for the isolation of SC and HC, respectively. Collected proteins are measured by NanoDrop and 2 μg aliquots are collected and digested by trypsin for proteomics analysis.

2. Hard corona. When using the fishing method, the authors perform an incubation of NPs in 10% FBS at 2-time points (3 minutes and 30 minutes), identifying by LC-MS/MS 262 (3 min) and 217 (30 minutes) proteins for D-NPS and 255 (3 min) and 197 (30 minutes) for L-NPs. When using the centrifugation method (to make comparisons between the 2 approaches), the authors perform an incubation in 10% serum using only one-time point (1 h at 37 degrees), different from those used in the fishing approach. This is not appropriate. The same exact conditions must be used when comparing the two approaches. In addition, what does 10% serum mean? Is it 10% FBS? If so, please use the same language. Moreover, at which temperature the Fishing experiment has been performed? Temperature is another important parameter and must be the same. The results of the centrifugation experiments for HC formation performed in the same conditions of the fishing approach are required.

Response. Thanks for your useful comment and suggestion.

1) Yes, it is not appropriate that the data for hard corona proteins by centrifugation (HCC) at 1 h incubation were compared with the data for SC and HC at 3 min and 30 min *via our* Fishing approach. Thus, we have followed the suggestion and have conducted a new set of experiments with 10% mouse serum upon 3 min and 30 min incubation and replaced the HCC data.

The updated data:

- Protein composition and abundance for 10% mouse serum: *Figs. 2B, 2C, Supplementary Fig. 4*
- HCC composition and protein abundance at 3 min and 30 min: *Figs. 2B, 2D-2G; Supplementary Figs. 4, 5*
- Comparison of HCC with HC: *Supplementary Figs. 6-7*
- Proteomics data are available in Supplementary Data in the section titled file and ProteomeXchange Consortium.
- Main text: HCC result and discussion (Pages 8-9 *Identification of hard corona composition after centrifugation*).

2) To be consistent, we have replaced the previous description (such as 10% serum or 10% FBS) to 10% mouse serum in the revised manuscript.

3) Experimental temperature: Both “Fishing” and centrifugation experiments were done at 37°C. We have added several sentences on Page 21, “To better mimic physiological environment, the Fishing experiments were run under a continuous shaking of sample plate (1,000 × rpm) at 37 °C that ensured the homogenous concentration of serum proteins in the sample solution during corona formation.”

3. Figure 2 and 3. The pie charts should have the same color legends (*e.g.* albumin pink/grey)

Response. Thank you. In the revised manuscript, we have presented the same proteins in different pie charts with the same colors (*Figs. 2, 3, and 4*).

4. Soft corona. When using the fishing method for soft corona formation, the authors use 2-time points (3 min and 30 min) identifying by LC-MS/MS 394 (3 min) and 288 (30 minutes) proteins for D-NPS and 424 (3 min) and 324 (30 minutes) for L-NPs. As expected, most of

the identified proteins (90%) are common to those identified in the HC proteins, thus demonstrating that the different concentration of washing buffer is not a valid method to separately elute soft and hard corona.

Response. Thanks for your constructive comment. According to previous LC-MS/MS data, the similarity of HC and SC composition was high, ~90%, however, the abundance of main proteins including albumin and immunoglobulins were significantly different between HC and SC upon 3 and 30 min incubation (*Fig. R1*). **We thus think that the washing buffer at different concentrations is valid in eluting HC and SC separately.**

Fig. R1. Abundance in the main SC and HC proteins formed on the surface of chiral NPs after 3 min (A) and 30 min (B) incubation with 10% FBS.

We have done new experiments for 10% mouse serum and the results clearly indicated the feasibility of WB at different concentrations for this study. The evidences are listed as below:

1) Technically, BLI data clearly showed that SC and HC can be eluted by 0.005% and 0.1% of WB, respectively (Supplementary Figs. 2 and 3). Based on BLI sensors, the wavelength shift indicates the change in bio-layer thickness during the adsorption and

desorption of proteins on the NPs. For the adsorption process, the wavelength shift increased with time and then reached an equilibrium (*Supplementary Fig.2*). For the elution process, the wavelength shift correlated to the decrease in protein amount on the NPs. Compared with Equilibration state, the shift of Baseline curve reflected the protein amount remained on the NPs. We subsequently observed that the proteins were partly eluted by 0.005%, 0.01%, and 0.05% WB and fully washed out by 0.1% WB and finally back to Baseline. Then, 0.005% and 0.1% of WB are chosen for the isolation of SC and HC, respectively (*Supplementary Fig. 3*). Thus, we verified the successful elution of SC and HC by 0.005% of WB and 0.1% of WB, respectively.

2) Experimentally, we have performed new experiments for corona study in 10% mouse serum and found a similarity of ~70% in HC and SC composition (*Supplementary Figs. 10E-H*), however, the abundance of HC and SC composition is quite different (*Fig. R2, or Figs. 3C, D; 4C, D; Supplementary Fig. 11*). In the new experiments, washing buffers can efficiently elute the HC and SC layers separately that well explains that **HC and SC show similar composition but different protein abundance**. Importantly, the similar as our work, other studies (*Response Refs. 5, 6*) also supported that the SC proteins are present also in the HC.

Figure 3D

Coronas of L-NPs

Figure 4D

- Albumin
- Others
- Complements
- Lipoproteins
- Acute Phase
- Immunoglobulins

Fig. R2. Proteomic characterization of time-dependent HC (Fig. 3D) and SC (Fig. 4D) components on L-NPs. At 3 min, the abundance of immunoglobins in HC is 3.3% (Fig. 3D), which is much lower than that (17.7%) in SC (Fig. 4D). Except of immunoglobins, the abundance of albumin and other components in HC and SC shows slight difference. At 30 min, the abundance of immunoglobins in HC is 29.5% (Fig. 3D), which is much lower than that (3.6%) in SC (Fig. 4D). The abundance of acute phase proteins in HC and SC is 34.5% and 23.6%, which is quite distinct. Moreover, the abundance of albumin and other components in HC shows significantly different from SC: albumin (11.4% vs 23.3%), and other components (21.7% vs 46.2%).

Supplementary Figure 11. Heatmap of the SC vs HC composition on chiral NPs at different incubation time. (A, B) Comparison in HC and SC components on D-NPs (A) and L-NPs (B) at 3 min and 30 min incubation. (C, D) Comparison in surface chirality-specific HC and SC components on NPs at 3 min (C) and 30 min (D) incubation. Corona components are isolated by the BLI-based “Fishing” method. Heatmaps show the median normalized abundance of protein groups detected for HC and SC protein composition

(columns) on chiral NPs (rows). Hierarchical clustering is performed in Proteome Discoverer 2.4. All data are acquired from three biological replicates.

In details, the similarity in coronal composition between HC and SC was about 69% for D-NPs and 74% for L-NPs at 3 min, while the similarity between HC and SC remained 75% for D-NPs and 68% for L-NPs at 30 min (*Supplementary Figs. 10E-H*). **Although the same composition of multiple proteins in HC exist as SC composition, their abundance was quite different** (*Supplementary Fig. 11; Figs. 3C, 3D, 4C, 4D*) **due to the competitive binding and exchange of various proteins with different binding affinity**. Herein, we took the abundance of HC and SC proteins on L-NPs for an example (*Fig. R2 or Figs. 3D, 4D*). At 3 min, the abundance of immunoglobins in HC is much lower than that in SC, while the abundance of albumin and other components in HC and SC shows slight difference. At 30 min, the abundance of immunoglobins, acute phase, albumin and other components in HC showed significantly different from that in SC.

3) Mechanism of corona formation: We explain why HC and SC show similar protein identities but distinct abundance of proteins in the following reasons. One of the reasons is that protein adsorption and exchange are a thermodynamic and kinetics process during protein corona formation (*Response Refs. 1, 2*). When the adsorption begins in physiological fluids, the proteins with high abundance but low diffusion rates can be immediately adsorbed on the surface of NMs as soft corona or the inner layer corona (HC). With prolonging incubation time, competitive adsorption of various proteins with higher binding affinities drives the exchange of proteins that changes the amounts of protein components at the inner layers, which is kinetics process for protein adsorption (*Response Refs. 1-3, 5,6*). A part of HC composition at inner layer will be gradually replaced by the proteins at outer layer overtime, suggesting the diffusion and competition replacement of proteins with lower affinity by those with higher affinity. Protein corona formation and evolution are thus spatiotemporal processes companying with bi-directional movement of various proteins, which suggests the appearance of protein composition in both HC and SC. **Our study had revealed the phenomenon that the compositions** (*Supplementary Figs. 10E-H*) **of SC and HC seemed to be similar, but the abundance of proteins** (*Figs. 3C,D; 4C,D*) **in HC and**

SC are quite distinct, which was supported by other studies (*Response Refs. 5, 6*). Therefore, we concluded that the interaction of NPs and protein, as well as the interplay among neighboring proteins contribute largely to corona formation and evolution, which can be affected by the factors including their abundance, thermodynamic diffusion, and binding affinities.

Therefore, we have updated contents in the revised manuscript on Page 7, “**To better address the challenges, we divided the multilayers of corona into outer and inner sections, i.e., the soft and hard coronas in this study.**”

In 10% mouse serum system, the similarity of HC and SC decreased from ~90% to ~70% compared to previous 10% FBS results. We have modified the description in the revised manuscript on Page 11: “**High similarity in coronal composition between HC and SC was found on two chiral NPs (>68%) at 3 and 30 min (Supplementary Figs. 10E-H). Although the same proteins can simultaneously exist as soft and hard corona composition, the abundance of HC and SC proteins was obviously different at 3 and 30 min (Supplementary Figs. 11A, B)**”

Response References.

- (1) Latreille, P. L.; Le Goas, M.; Salimi, S.; Robert, J.; De Crescenzo, G.; Boffito, D. C.; Martinez, V. A.; Hildgen, P.; Banquy, X., Scratching the Surface of the Protein Corona: Challenging Measurements and Controversies. *ACS Nano* **2022**, *16* (2), 1689-1707.
- (2) Zhdanov, V. P.; Cho, N. J., Kinetics of the formation of a protein corona around nanoparticles. *Math Biosci.* **2016**, *282*, 82-90.
- (3) Vilanova, O.; Mittag, J. J.; Kelly, P. M.; Milani, S.; Dawson, K. A.; Radler, J. O.; Franzese, G., Understanding the Kinetics of Protein-Nanoparticle Corona Formation. *ACS Nano* **2016**, *10* (12), 10842-10850. (4) Zhang, Y. W.; Wu, J. L. Y.; Lazarovits, J.; Chan, W. C. W., An Analysis of the Binding Function and Structural Organization of the Protein Corona. *J. Am. Chem. Soc.* **2020**, *142* (19), 8827-8836.
- (5) Mohammad-Beigi, H. *et al.* Mapping and identification of soft corona proteins at nanoparticles and their impact on cellular association. *Nat. Commun.* **2020**, *11*, 4535. Cited as Ref. 19 in the revised manuscript.

(6) Weber, C., Simon, J., Mailänder, V., Morsbach, S., Landfester, K. Preservation of the soft protein corona in distinct flow allows identification of weakly bound proteins. *Acta Biomater.* **2018**, *76*, 217-224. Cited as Ref. 21 in the revised manuscript.

5. The authors should consider revising the entire work from another angle: using the time (3 minutes vs 30 minutes) as a factor for discriminating soft and hard corona. Indeed, it is well established that the soft corona is formed in the very early stage of PC formation and that later this is replaced by the hard corona. (see Tenzer S. *et al Nat Nano* 2013)

Response. Yes. We totally agree that time is a crucial factor for protein corona (PC) formation and evolution that was also mentioned in the publication by Tenzer S. et al (*Nat Nano* 2013). It is a very nice and breakthrough work which has shed light on the latter research that has been cited as Ref. 48. In our current study, differently, **we would like to focus that the formation and evolution of SC and HC is a spatiotemporal process (time-dependent adsorption and exchange) together with dynamic change in the structures (multilayers of proteins, varied composition and abundance of proteins), which was supported by previous publications (Response Refs. 1-6) and our results.**

In details, at the beginning, proteins with high abundance but low diffusion rates can be adsorbed on the surface of NPs and they immediately form the inner layer of protein corona, which is a thermodynamic process, as also indicated by other work (*Response Ref. 1, 2*). With prolonging time, proteins with higher affinities bind to the surface of NPs due to competitive interaction and exchange as a kinetics process, which was supported by previous publications (*Response Refs. 1,2,3,5-6*) and our results. A part of coronal composition at the inner layer will be gradually replaced by proteins at the outer layer overtime, due to the diffusion of proteins and competitive replacement among various proteins. Moreover, the abundance of proteins with high affinity at the inner layer will increase with time until the adsorption equilibrium. Therefore, protein corona formation and evolution are a continuous and dynamic process, during which bi-directional movement of various proteins with distinct affinities and abundances persistently happens.

In this work, we have done new experiments to prove the spatiotemporal process of SC and HC formation and evolution. We divided protein corona into two layers: the outer one as SC and the inner together with one as HC based on the spatial location of coronal

composition. The evidences show that coronal structure including the composition and the abundance proteins changed with time: (1) **The composition in both HC and SC varies overtime** (*Supplementary Fig. 10C, D; Fig. 10I, J*). (2) **The abundance of HC and SC composition also changes overtime due to distinct affinities of various proteins binding to either NPs or the HC and competitive replacement** (*Figs. 3C, D; 4C, D, Supplementary Figs. 8, 9, 11*). We took immunoglobins and albumin for examples. Comparably, immunoglobins exhibit much higher affinity to NPs (*Figs. 5B, C*) and to pre-coated serum HC (*Figs. 5E, F*), suggesting that immunoglobins will gradually become a major component in both HC and SC than albumin, which was supported by the data of enrichment factors (*Supplementary Fig. 12*).

In conclusion, we have considered both time and structure factors for the study of protein corona and developed the strategy to isolate proteins from the outer layer to the inner one for HC and SC proteomics at different incubation time. We believe that this study can help comprehensively understand the formation and evolution of protein corona.

6. Despite much information could be obtained from this work, I don't see a great novelty in it. The authors should emphasize the novelty of such work if they aim to publish it in a high IF journal.

Response. Thanks for this reviewer's suggestion. We have emphasized the novelty mainly in two aspects: (1) With respect to the methodology, the "Fishing" strategy is quite novel that successfully isolates and identifies hard and soft corona *in situ* for the first time and shows its great advantages compared to conventional methods. (2) With respect to biological effects of SC, this work would be the first report that reveals how chiral surface regulates SC formation and subsequent trafficking or fates of NPs *in vivo*. As below, we list the novelty of the work in details.

Firstly, this "Fishing" strategy is novel and powerful to monitor and characterize protein corona formation and evolution *in situ* rather than a time-consuming isolation and centrifugation step. To study protein corona, centrifugation-based method is the most common and popular way to separate and collect coronal proteins from biological fluids. **However, centrifugation-based method has the certain limitations, *i.e.*, unable to collect soft corona, unsuitable for ultra-small nanoparticles which are hard to spin down by**

centrifugation, time consuming and unappreciated for dynamic analysis. Comparably, BLI is a real-time, dynamic technique for intermolecular affinity detection, which is developed recently as a state-of-the art technique (*Cell*, 2020, 181(2): 281-292; *Nat. Biomed. Eng.*, 2021, 5(11): 1288-1305). Herein, BLI-based “Fishing” method is time-saving and appropriate to characterize protein adsorption and desorption *in situ* that can study dynamic process of protein corona formation and evolution ranging from several seconds/minutes to hours. In addition, this method can *in situ* separate the corona on specific nanomaterials such as quantum dots, ultra-small NPs, ultrathin 2D nanosheets, and soft nanomaterials with low density that are difficult/hardly to be centrifugated and isolated from biological fluids in short time.

Importantly, this BLI-based “Fishing” method is powerful to reveal how protein corona mediates the fast clearance of NMs by monocyte-macrophage systems. As we know, most NMs can be quickly cleared by monocyte-macrophage systems, but little is known about early events of adsorption due to methodological challenge. In this work, we developed this method to study fast formation and evolution of protein corona on ultra-small Cu₂S NPs at 3 and 30 min. As a conventional way, centrifugation-based method costs 1 hour to isolate NPs-protein complex from biological fluids, however, it fails to characterize the component of protein corona at very early time (several seconds to minutes post-incubation). Comparably, BLI-based method can be directly used to isolate coronal components at 3 min or ideally even several seconds before MS identification and quantification. Therefore, the novel method is excellent to realize both simple, time-saving, dynamic study for protein corona, and *in situ* study about the corona on many types of NMs that cannot be easily and fast isolated.

Secondly, we offer a simple and *in situ* way to characterize and quantify the components in multilayers of protein corona. Current methods for corona analysis primarily rely on centrifugation separation, which requires the isolation of NM-corona complexes from biological fluids *via* high-speed centrifugation and buffer rinses for several times. However, only HC proteins are retained during centrifugation, and SC components are lost. Differently, our study presented a direct way to real time monitor and *in situ* separate protein coronas in biological fluids for LC-MS/MS identification, allowing us to perform time- and surface-dependent analysis of protein components at the outer and inner layers.

Thirdly, we reveal that surface chirality-specific soft corona regulates the ADME behaviors of NPs *in vivo*. Surface chirality is one of important properties of NPs that influences biological responses to NPs and potential biomedical applications. However, little is known about whether surface chirality is involved in the trafficking, metabolism, and clearance of NPs *in vivo*. In this work, we found that L- and D-Cu₂S NPs exhibit distinct rates for blood clearance and liver/spleen metabolism. We further quantified both HC and SC composition and components and found that two NPs show distinct enrichments of immunoglobins in SC at 30 min which well explained why D-NPs have a fast blood clearance. This work is the first example about how surface chirality-specific SC mediates *in vivo* fates of NPs, which will be one of leading studies about the roles of SC in regulating the fates of NPs and biological effects.

Taking together, we believe the above discussion clearly addresses the novelty of this work. In the revised manuscript, we have provided additional discussion as below:

In Introduction on Page 5, “Compared to the centrifugation-based method, this integrated strategy is powerful to study formation and evolution of protein corona that allows real-time monitoring of protein adsorption and dissociation and supports in situ and fast isolation of multiple layer corona i.e., HC and SC for accurate identification and quantification of protein components. The BLI-based “Fishing” method is also feasible to perform corona study in complex biological fluids without further centrifugation and in dynamic and very fast interaction systems for protein corona especially for ultra-small NPs, ultrathin nanosheets, and NMs with low density that are hardly to be centrifugated in seconds and several minutes.”

In the section of Result-/ *Effect of surface chirality on biodistribution of NPs in vivo and its correlation with SC* on Pages 16-17, “Surface chirality is one important property of NMs, however, little is known about whether surface chirality mediates the trafficking, metabolism, and clearance of NMs *in vivo*. In this work, BLI-based “Fishing” method was used to reveal chirality-characteristic HC and SC components and dynamic change with time. Two chiral NPs showed distinct enrichments of immunoglobins in SC which can well explain why D-NPs have a fast blood clearance rate. In conclusion, surface chirality determines the formation and evolution of specific SC that mediates *in vivo* ADME fates of NMs and shows the importance role of SC in regulating the fates of NMs and biological effects.”

In the section of Summary on Page 18, “Importantly, the “Fishing” strategy can be extended to a simple and universal method to study how physicochemical properties of NPs (including elemental composition, surface charges and modification, surface roughness, and size *etc.*) affect the formation of HC/SC and biological responses. For example, BLI-based “Fishing” method can be used to *in situ* characterize corona formation on specific nanomaterials such as quantum dots, ultra-small NPs, ultrathin 2D nanosheet, soft nanomaterials with low density that are difficult to be centrifugated to isolate NPs-corona from protein solution. Moreover, this approach will offer opportunities to study different coronas for nanomaterials in various biological fluids/microenvironments (plasma, cerebrospinal fluid, pulmonary fluid, gastro-intestinal fluid, *etc.*). In future works, dynamic exchange of protein coronas in different biological sources could be also investigated that helps deeply understand corona evolution during transfer among different fluids and is powerful for the study of biological effects of nanomaterials and the developments of nanomedicines.”

Reviewer #3

The outer layer of protein coronas, *i.e.*, soft corona contributes to “biological” identity of NPs that regulates fundamental effects and fates of NPs *in vivo* and *in vitro*, however, little is known about the soft corona composition due to a longstanding challenge in soft corona analysis. This work made a breakthrough in soft corona analysis, in which authors presented a simple method to precisely capture several layers of proteins on NPs in complex biological matrix or media. Combining BLI-biosensor with MS, the strategy can monitor the formation and exchange of protein corona in real time that helps fast isolation of proteins at outer and inner layers and MS analysis to identify protein composition. This study is interesting and has potential broad applications in multidisciplinary fields such as nanobiology, nanomedicine, biophysics, and environment science, *etc.* I would like to recommend its publication after minor revision.

1. The “Fishing” strategy is useful in studying the relationship between surface properties of NPs and biological responses due to corona formation. Besides chirality, authors may point out more applications in the discussion to explain the importance of the method.

Response. Thanks for the positive comments and suggestion. Besides the use in exploring chiral effect on corona formation, we discussed about the application of the “Fishing” strategy in more systems as a simple, universal, and fast method. These systems include the corona study of NPs with various physicochemical properties and within different biological fluids/microenvironments. We have added the discussion in the manuscript Page 18.

“Importantly, the “Fishing” strategy can be extended to a simple and universal method to study how physicochemical properties of NPs (including elemental composition, surface charges and modification, surface roughness, and size *etc.*) affect the formation of HC/SC and biological responses. For example, BLI-based “Fishing” method can be used to *in situ* characterize corona formation on specific nanomaterials such as quantum dots, ultra-small NPs, ultrathin 2D nanosheet, soft nanomaterials with low density that are difficult to be centrifugated to isolate NPs-corona from protein solution. Moreover, this approach will offer opportunities to study different coronas for nanomaterials in various biological fluids/microenvironments (plasma, cerebro-spinal fluid, pulmonary fluid, gastro-intestinal fluid, *etc.*).”

2. About the novelty of method, it is better to compare this method more with other methods in discussion or in the introduction.

Response. Thanks for this suggestion. Previously, we made some comparison of this method with others in Introduction's second paragraph "To characterize SC components, several analysis methods have been developed.....". To emphasize this method's novelty in the revision, we have added several sentences in the Introduction Page 5, "Compared to the centrifugation-based method, this integrated strategy is powerful to study formation and evolution of protein corona that allows real-time monitoring of protein adsorption and dissociation and supports in situ and fast isolation of multiple layer corona i.e., HC and SC for accurate identification and quantification of protein components. The BLI-based "Fishing" method is also feasible to perform corona study in complex biological fluids without further centrifugation and in dynamic and very fast interaction systems for protein corona especially for ultra-small NPs, ultrathin nanosheets, and NMs with low density that are hardly to be centrifugated in seconds and several minutes."

3. Check and write down proper abbreviation and full names such as BLI.

Response. Thanks. Full names of all abbreviations have been checked and updated including nanoparticles (NPs), Bio-Layer Interferometry (BLI), cuprous sulfide (Cu₂S), adsorption, distribution, metabolism, and excretion (ADME), X-ray diffraction (XRD), and Liquid Chromatography-Tandem Mass Spectrometry (LC-MS/MS).

4. Chiral molecules are close to the adsorbed HC proteins and may affect the formation of HC. Authors can discuss more about chirality-based HC protein "fingerprint" about the type subtypes.

Response. Thanks. We have discussed more about the chirality-based HC protein "fingerprint" in the revision.

According to Supplementary Figs. 8 and 9, 11C, D (as below), we observe that HC protein "fingerprint", i.e., the characteristic components of HC on both types of NPs, changed with chirality and time. At 3 min incubation, major HC components for L-NPs include: Others, Albumin > Acute Phase proteins > Complements, Immunoglobulin > Lipoproteins, while those for D-NPs include: Others, Albumin > Immunoglobulin > Complements,

Lipoproteins. The abundances in two NPs' HC component at 3 min exhibited obvious difference. L-NPs preferred to adsorb more HC components such as complements (P06909, P01027) and acute phase proteins (P31532) as well as some subtypes of lipoproteins (P06728) than D-NPs, meantime, D-NPs preferred to adsorb some subtypes of lipoproteins (Q6LD55) (Supplementary Fig. 8). At 30 min incubation, the differences in HC components for two chiral NPs changed with time. Major components of HC for both NPs showed similar results: Others, Immunoglobulin, Acute Phase proteins > Albumin > Complement > Lipoproteins. For HC components, L-NPs preferred to adsorb much less albumin than D-NPs. L-NPs preferred to adsorb more Immunoglobulin (A0A0F7QZE4), Acute Phase protein (P31532), complement (P01027), and lipoprotein (Q6LD55, P34928).

Thus, we demonstrated that the abundance of HC proteins such as Immunoglobulins, Acute Phase proteins, Albumin, and the subtypes of them vary with chiral surface and incubation time, which can be regarded as dynamic fingerprints for chiral NPs' HC. We would like to emphasize that it is difficult to reach a conclusion for the whole picture about the characteristic proteins for chiral surface at different incubation time.

Supplementary Fig. 11 (C, D) Comparison of surface chirality-specific HC with SC components on NPs at 3 min (C) and 30 min (D) incubation. Corona components are isolated by the BLI-based “Fishing” method. Heatmaps show the median normalized abundance of protein groups detected from the coronal proteins (rows) on chiral NPs (columns).

Detailed discussion, please refer to the revised main text (on Pages 10-11 and Page 17): “We next studied the effect of surface chirality on HC properties, including composition and abundance. After 3 min of incubation (Fig. 3C, Supplementary Fig. 8), the relative abundance of acute phase proteins on both chiral NPs showed significant difference (~0.8% for D-NPs vs ~16% for L-NPs). Moreover, chirality-specific difference in HC protein abundance was ~6.2% for albumin and ~10.5% for other proteins. Interestingly, after 30 min incubation (Fig. 3C, Supplementary Fig. 9), the significant difference in HC protein abundance on both NPs was observed for albumin (~6.1%), others (~16.1%), acute phase (~9.8%), and immunoglobulin (~11.0%). Similar trend in HC abundance for chiral NPs was that albumin and other proteins decreased, while acute phase and immunoglobulins increased over time. Collectively, the chirality influences HC composition and abundance at the early stage with significant changes over time (Supplementary Figs. 10A, B), highlighting the dynamic effect of chirality (Supplementary Figs. 10C, D) on HC formation, i.e., dynamic fingerprints for chiral NPs’ HC. Herein, we emphasize that both time and surface chirality of NPs play important roles in the formation and evolution of hard corona.”

“We believe that the collective coronal fingerprint at the nano-bio interface determines the distribution, clearance, and coronal remodeling in the blood. Similarly, the enrichment of acute phase and immunoglobulins on NMs, shortened blood circulation time and enhanced accumulations in organs, such as liver and kidney.⁴⁹ We assumed that the surface properties of NMs are essential to regulate dynamic evolution of HC at early stage, that further determines its SC composition. Therefore, these dynamic coronal exchanges are critical for in vivo recognitions. Interestingly, we observed that protein corona formation and remodeling is a stereoselective process, which enhances over time, however, further studies are required. It should be noted that coronal evolution might depend on physiological fluid conditions, size, shape, surface properties and surface coatings of NPs. The relationship of physicochemical properties with HC and SC components will help predict biological fates of NMs and responses that paves a way for the rational design and application of nanomedicine.”

5. They used wash buffer to elute the proteins at outer layer and inner layer. They may show BLI data about the adsorption and desorption of NPs on the sensors to show the stability of NPs during the wash step.

Response. Thanks for this suggestion. In Supplementary Fig. 2, the shift ($\Delta\lambda$) difference of the final baseline and equilibrium steps indicated that the chiral NPs remain on the sensor before and after introducing to biofluid and washing buffers. In details, the NPs remained on the sensor during the loading step. During the adsorption process, the adsorption of proteins increased with time to reach an equilibrium. During the washing step, the protein corona of NPs for the control (washed by pure water) remained on the sensor, while WB at a series of concentrations enables the protein corona to be washed out from the surface of NPs at different degrees. It's worth noting that the signals for NP-protein complex in WB did not decrease at the final baseline compared to the signals at the equilibrium curve, suggesting a stable condition of NPs on the sensor.

In addition, we added the discussion on Page 8, “Therefore, WB at present concentrations could elute multilayered proteins from the surface of NPs with the immobilized NPs on the sensors during the washing (Supplementary Figs. 2, 3).”

REVIEWERS' COMMENTS

Reviewer #1 (Remarks to the Author):

The authors have considered my concerns in the revised manuscript; the outcomes of the manuscript, with all essential controls and experimental details, are now very robust and can be reproduced by other researchers.

Responses to the Reviewers

Remarks to the Author: Reviewer #1:

The authors have considered my concerns in the revised manuscript; the outcomes of the manuscript, with all essential controls and experimental details, are now very robust and can be reproduced by other researchers.

Reply: We appreciate this referee for this positive comment.